# Heterogeneous presynaptic receptive fields contribute to directional tuning in starburst amacrine cells

John A Gaynes, Samuel A Budoff, Michael J Grybko, Alon Poleg-Polsky*

Department of Physiology and Biophysics, University of Colorado School of Medicine, Aurora, United States

*For correspondence: Alon.Poleg-Polsky@cuanschutz. edu

Competing interest: The authors declare that no competing interests exist.

**Abstract** The processing of visual information by retinal starburst amacrine cells (SACs) involves transforming excitatory input from bipolar cells (BCs) into directional calcium output. While previous studies have suggested that an asymmetry in the kinetic properties of BCs along the soma-dendritic axes of the postsynaptic cell could enhance directional tuning at the level of individual branches, it remains unclear whether biologically relevant presynaptic kinetics contribute to direction selectivity (DS) when visual stimulation engages the entire dendritic tree. To address this question, we built multicompartmental models of the bipolar–SAC circuit and trained them to boost directional tuning. We report that despite significant dendritic crosstalk and dissimilar directional preferences along the dendrites that occur during whole-cell stimulation, the rules that guide BC kinetics leading to optimal DS are similar to the single-dendrite condition. To correlate model predictions to empirical findings, we utilized two-photon glutamate imaging to study the dynamics of bipolar release onto ON- and OFF-starburst dendrites in the murine retina. We reveal diverse presynaptic dynamics in response to motion in both BC populations; algorithms trained on the experimental data suggested that the differences in the temporal release kinetics are likely to correspond to heterogeneous receptive field properties among the different BC types, including the spatial extent of the center and surround components. In addition, we demonstrate that circuit architecture composed of presynaptic units with experimentally recorded dynamics could enhance directional drive but not to levels that replicate empirical findings, suggesting other DS mechanisms are required to explain SAC function. Our study provides new insights into the complex mechanisms underlying DS in retinal processing and highlights the potential contribution of presynaptic kinetics to the computation of visual information by SACs.

## eLife assessment

This **important** study uses a combination of computational modeling and glutamate imaging to show how a particular synaptic organization referred to as space-time wiring contributes minimally to a dendritic computation that occurs in the retina. The evidence supporting the claims of the authors is **compelling**, incorporating new findings regarding dynamic receptive field properties, an improvement over previous modeling and experimental results based on static visual stimuli. The work will be of interest to retinal neurobiologists and neurophysiologists interested in dendritic computations.

## Introduction

Starburst amacrine cells (SACs) play a crucial and well-established role in the computation of direction selectivity (DS) in the mammalian retina (**Demb, 2007**; **Mauss et al., 2017**; **Wei, 2018**). SACs can be

broadly categorized into two subtypes: ON and OFF cells, both of which receive excitatory inputs from diverse populations of bipolar cells (BCs). In mice, the majority of input synapses onto SACs are concentrated in the proximal two-thirds of the radially symmetric dendritic tree, while the distal dendrites house the output synapses (*Ding et al., 2016*; *Vlasits et al., 2016*). Individual SAC branches exhibit robust calcium responses to objects moving in the outward direction (away from the soma); consequently, different parts of the dendritic tree demonstrate distinct directional preferences (*Chen et al., 2016*; *Ding et al., 2016*; *Euler et al., 2002*; *Koren et al., 2017*; *Morrie and Feller, 2018*; *Poleg-Polsky and Diamond, 2016*; *Poleg-Polsky et al., 2018*). SACs are essential for generating DS in direction-selective ganglion cells (DSGCs); they provide a combination of cholinergic excitation and GABAergic inhibition that specifically target co-stratifying DSGCs sharing the directional preference of the releasing site (*Briggman et al., 2011*; *Fried et al., 2002*; *Kostadinov and Sanes, 2015*; *Lee et al., 2010*; *Morrie and Feller, 2015*; *Pei et al., 2015*; *Sethuramanujam et al., 2021*; *Soto et al., 2019*; *Wei et al., 2011*; *Yonehara et al., 2011*; *Yonehara et al., 2016*).

Numerous mechanisms have been proposed to contribute to the establishment of DS in SACs. For example, network mechanisms, such as feedback inhibition onto neighboring SACs or SAC–derived cholinergic excitation of presynaptic BCs, were shown to enhance DS (*Ding et al., 2016*; *Hellmer et al., 2021*; *Münch and Werblin, 2006*; *Poleg-Polsky et al., 2018*). However, it is worth noting that SACs maintain their directional tuning in the absence of cholinergic and GABAergic transmission (*Chen et al., 2016*), indicating that other mechanisms must contribute to directional tuning. Other proposals emphasized the involvement of cell-autonomous processes that encompass dendritic voltage gradients sustained by tonic excitation from BCs (*Gavrikov et al., 2003*; *Hausselt et al., 2007*) and fine-tuned morphology that imposes membrane filtering of postsynaptic signals (*Tukker et al., 2004*; *Vlasits et al., 2016*). While these mechanisms can contribute to the development of DS to some degree, they fall short in predicting the experimentally observed levels of directional tuning. In addition, it remains unclear whether these requirements for tonically active presynaptic release described in the rabbit are fulfilled in the mouse circuit, and cells with abnormal shapes still exhibit near-normal levels of DS (*Morrie and Feller, 2018*).

Recently, a reformulation of the classical Hassenstein–Reichardt correlator model has been proposed, which emphasizes the kinetic properties of glutamatergic inputs sampled by SAC dendrites as contributing factors to DS (*Greene et al., 2016*; *Kim et al., 2014*). Known as the 'space-time wiring' model, this formulation is based on connectomic reconstructions of bipolar–SAC contacts. Both ON- and OFF-SAC dendrites tend to stratify closer to the middle of the inner plexiform layer (IPL) as they branch out from the soma. This unique stratification pattern enables SAC dendrites to receive inputs from different types of BCs in their proximal and distal regions (*Ding et al., 2016*; *Greene et al., 2016*; *Kim et al., 2014*). Importantly, functional studies have uncovered a positive correlation between the transiency of BC axonal release and the distance from the boundaries of the IPL (*Baden et al., 2013*; *Chen et al., 2014*; *Euler et al., 2014*; *Franke et al., 2017*). Specifically, BC types that stratify closer to SAC somas (OFF-SACs: BC1-2; ON-SACs: BC7) exhibit prolonged release, whereas more distal types (OFF-SACs: BC3-4; ON-SACs: BC5) tend to respond with a transient pattern. According to the space-time wiring hypothesis, these regional differences facilitate a more effective summation of excitatory inputs in the outward direction.

Two main methodologies were taken to examine the existence of the space-time wiring hypothesis in the bipolar–SAC circuit. First, with the caveat that the long, thin dendritic processes in SACs impose significant challenges for an effective space clamp (*Poleg-Polsky and Diamond, 2011*; *Stincic et al., 2016*), it is possible to measure the distance dependence of BC dynamics by analyzing somatic excitatory postsynaptic currents. Although one study reported substantial differences in the shape of the recorded currents, primarily in the OFF-SACs (*Fransen and Borghuis, 2017*), a second group that recorded from ON-SACs failed to replicate this finding (*Stincic et al., 2016*).

A second method utilized genetically encoded fluorescent glutamate sensors, such as iGluSnFR, to directly study the dynamics of BC release (*Borghuis et al., 2013*; *Marvin et al., 2013*). An examination of glutamate release dynamics confirmed the diverse signaling patterns within the lamina of the IPL accessible to SAC dendrites (*Franke et al., 2017*; *Gaynes et al., 2022*; *Strauss et al., 2022*). Furthermore, excitatory drive to ON-DSGCs, which share many glutamatergic inputs with ON-SACs, was found to consist of units with distinct release dynamics, theoretically supporting the type of directional computation proposed by the space-time wiring hypothesis (*Matsumoto et al., 2019*). Finally,

a project published last year described the asymmetric distribution of kinetically distinct iGluSnFR signals in ON-SACs (*Srivastava et al., 2022*).

However, as the investigation of BC signaling has predominantly focused on release dynamics to static stimulus presentations, the question arises: Can responses to flashed stimuli accurately represent the release waveforms produced by moving objects?

The receptive field (RF) engagement in response to a continuously moving stimulus follows a sequential pattern, with the surround being activated first, followed by the center. As a result, the motion response profile becomes a complex function, influenced by factors like RF size, temporal characteristics, and stimulation velocity (*Strauss et al., 2022*). This integration mode stands in stark contrast to the simultaneous activation of RF components observed during flashes of static objects. Indeed, our research has revealed a significant distinction in BC release between stationary and moving probes (*Gaynes et al., 2022*). This discrepancy emphasizes the limited capacity of flash responses to predict the fundamental characteristics of BC responses to motion accurately and highlights the importance of examining directional capabilities in SACs with physiologically relevant stimuli.

An additional crucial factor to consider is the dependence of the presynaptic response waveform with respect to the direction of activation. The space-time wiring model posits that BCs generate similar glutamatergic drive in both outward and inward directions. Yet, this assumption may be challenged when moving objects appear within the bipolar RF. Signal processing in such scenarios fundamentally differs from stimuli that are flashed or continuously moving, and these differences can vary significantly among different types of BCs (*Gaynes et al., 2022*; *Strauss et al., 2022*). The intricacies of signal transformation within the BC network give rise to important considerations regarding the application of advanced visual protocols utilized in previous studies to investigate the DS properties of SAC dendrites. These protocols were specifically designed to target a small patch of retina beneath a single SAC dendrite or induce circularly symmetric shrinking or expanding motion to elicit consistent responses across the dendritic tree (*Euler et al., 2002*; *Hausselt et al., 2007*; *Koren et al., 2017*; *Oesch and Taylor, 2010*). Nonlinear integration in BCs triggered by these visual stimuli may unintentionally introduce a directional bias to BC activation, contradicting the fundamental assumptions of the space-time wiring hypothesis (*Gaynes et al., 2022*).

Due to the challenges associated with examining the relevance of the space-time wiring model in the context of single dendritic computations, we opted instead to focus on understanding the directional performance of SACs when the entire dendritic tree is stimulated. Our approach involved the integration of machine-learning algorithms with multicompartmental models of the bipolar–SAC network. In addition, we employed glutamate imaging techniques to record the waveforms of excitatory synaptic drive experienced by ON- and OFF-SACs during object motion. We reconstructed the underlying RF structure that supports different bipolar release shapes and subsequently combined them into a detailed model. By doing so, we sought to gain insights into how diverse RF characteristics of BCs contribute to the directional performance observed in SAC, while also comparing them to the maximal theoretical capabilities of space-time models achieved with synthetic presynaptic RFs. Our findings indicate that the RF diversity observed in BCs generates response profiles that have the potential to contribute to SAC DS but to levels significantly below the maximum theoretical capabilities observed with synthetic RFs. Overall, by combining machine-learning techniques, multicompartmental modeling, and glutamate imaging, our study provides insights into the complex relationship between BCs and SACs in the context of DS.

## Results

### SAC morphology imposes substantial signal attenuation but does not preclude inter-branch interactions

Calcium signals occurring in the terminal dendrites of SACs exhibit remarkable DS, both when visual stimulation is focused on the recorded dendrite and when stimuli engage the entire dendritic tree (*Chen et al., 2016*; *Ding et al., 2016*; *Euler et al., 2002*; *Koren et al., 2017*; *Morrie and Feller, 2018*; *Poleg-Polsky and Diamond, 2016*; *Poleg-Polsky et al., 2018*).

Because the precise nature of SAC function when a stimulus encompasses the entire cell remains unclear, our first goal was to determine whether whole-cell activation should be understood as a combination of multiple independent processing units or as the sequential recruitment of interacting

dendrites. The distinctive dendritic morphology of SACs, characterized by long and narrow-diameter processes, was proposed to lead to the compartmentalization of individual or sister branches and significant signal attenuation to other dendrites (*Masland, 2005*; *Poleg-Polsky et al., 2018*). Yet theoretical studies suggested the presence of nontrivial dendritic crosstalk (*Ding et al., 2016*; *Tukker et al., 2004*).

To investigate whether the dendritic geometry and passive parameters support the hypothesis of independent computations in isolated dendritic compartments, we conducted a multicompartmental NEURON simulation to assess signal attenuation in reconstructed SACs. For our experiments, we selected a specific recording location on a terminal dendrite (*Figure 1A*, indicated in red) and administered 100-ms-long 10 pA current steps at various dendritic locations (*Figure 1A*). Consistent with prior studies, we observed a significant increase in signal attenuation as a function of the distance from the recording site (*Stincic et al., 2016*; *Tukker et al., 2004*; *Vlasits et al., 2016*). On average, approximately 40% of the injected signal was detectable at the recording location when the injection site was within the same branch, whereas stimulating other dendrites yielded negligible responses (*Figure 1A and B*).

By utilizing the built-in NEURON 'impedance' class to investigate signal propagation in relation to morphology, we observed a significant >20-fold attenuation on all branches distanced from the recording location by the soma. Notably, even sister dendrites sharing a common second-order bifurcation displayed a considerable >5-fold signal reduction (*Figure 1A*).

Subsequently, we examined whether dendritic isolation persists when multiple co-active inputs are present, which more accurately represents the activation pattern elicited by global motion. In mouse SACs, BCs innervate the proximal two-thirds of the dendritic tree (*Ding et al., 2016*; *Vlasits et al., 2016*). To replicate this arrangement, we distributed 200 synaptic inputs within a circle of 110 µm radius centered on the soma (*Figure 1C*). Each synapse exhibited a conductance of 0.1 nS and was stimulated by a 100-ms-long pulse.

Initially, we activated all the synapses that innervated dendrites sharing a common second-order branch. The peak voltage response following focal stimulation was measured at 10.9 ± 3.3 mV (mean ± SD, n = 100, *Figure 1C*). Next, we performed a similar manipulation, now stimulating excitatory synapses on the remaining dendritic tree (*Figure 1C*). Notably, the depolarization recorded from the same branch during the stimulation of other dendrites was higher, reaching 13.2 ± 1.9 mV (*Figure 1C*).

Hence, despite the substantial decrease in inter-dendrite signal propagation resulting from the cell's morphology and passive characteristics, the model predicts significant voltage interaction throughout the cell. This observation can be attributed to the disparity in the number of active inputs between the two conditions. Specifically, the combined length of dendrites sharing a common second-order parent constitutes less than 10% of the entire dendritic tree's length. This proportion also applies to the number of synapses involved in focal or global stimuli. Thus, although the efficacy of individual focal synapses is considerably more pronounced (*Figure 1A and B*), the larger number of inputs on other dendrites compensates for the signal attenuation caused by an individual synapse – leading to a comparable response to focal and global activation patterns (*Figure 1C*).

## Development of a machine-learning model for enhancing DS in BC–SAC interactions

Our findings highlight the existence of long-distance interactions between synaptic inputs that extend beyond the boundaries of a single dendritic compartment. Unless explicitly mentioned otherwise, all subsequent simulations and experiments described below employed full-field moving bars. This choice is physiologically relevant as SACs exhibit a high degree of DS in response to such stimuli (*Chen et al., 2016*; *Ding et al., 2016*; *Euler et al., 2002*; *Koren et al., 2017*; *Poleg-Polsky and Diamond, 2016*; *Poleg-Polsky et al., 2018*). Additionally, full-field stimulation circumvents the nonlinear processing of edges and emerging objects in BCs (*Gaynes et al., 2022*; *Strauss et al., 2022*), which could potentially influence the interpretation of responses to stimuli appearing within the SAC RF (see 'Discussion).

Previous studies suggested that differences in release profiles between proximal and distal BCs could contribute to directional tuning in isolated SAC dendrites, especially if proximal inputs exhibit more sustained and delayed dynamics (*Fransen and Borghuis, 2017*; *Greene et al., 2016*; *Kim et al., 2014*; *Srivastava et al., 2022*). However, do similar activation profiles promote DS during whole-cell

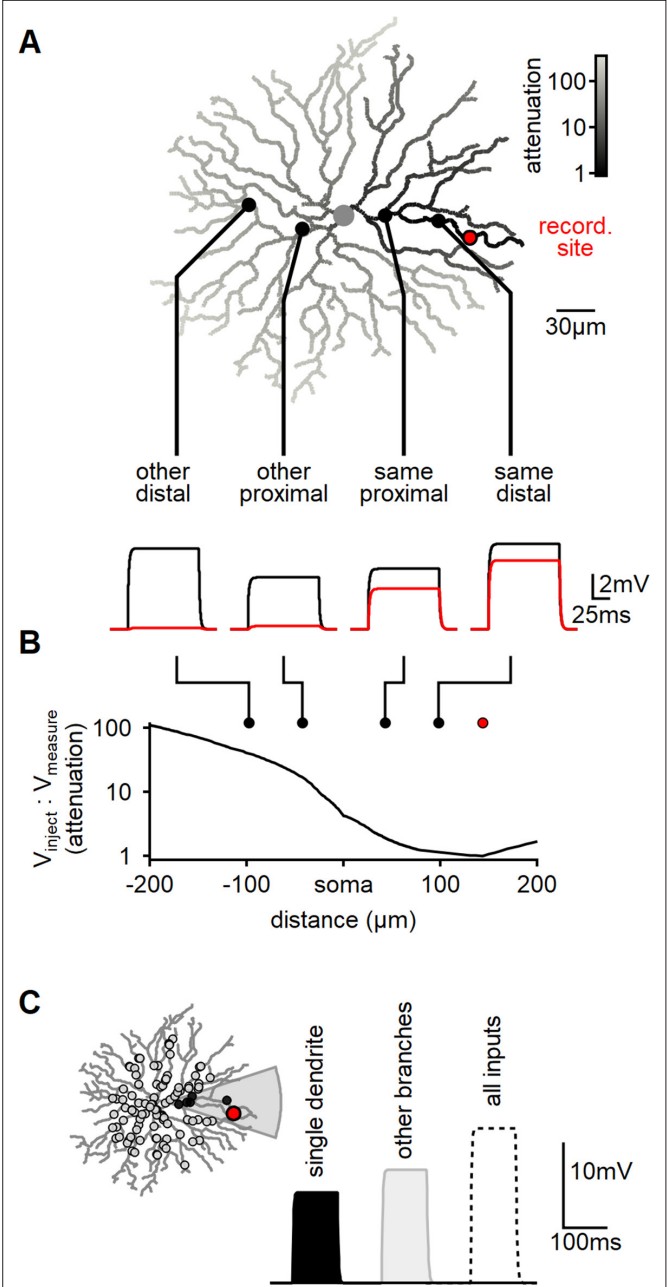

**Figure 1.** Modeling reveals substantial dendritic crosstalk in starburst amacrine cells (SACs) despite significant voltage attenuation. (**A**) Reconstructed morphology of a mouse SAC, with dendrites color-coded based on voltage attenuation toward the recording site (indicated in red). The bottom panel displays example voltage responses to a 100-ms-long current pulse injected at one of four positions denoted by circles. The black and red traces represent the potentials at the injection and recording sites, respectively. (**B**) Analysis of signal attenuation in SAC dendrites relative to the recording site marked in panel (**A**), as a function of distance from the soma (negative values indicate positions on the opposite side of the dendritic tree). (**C**) Comparison of peak depolarization resulting from synaptic stimulation of the recorded branch (labeled as 'single dendrite'; stimulated area shaded in gray) with the impact of driving synapses on other dendritic sites excluding the target branch (right, light gray). The combined activation of all inputs is shown for reference (dotted).

integration? To investigate the impact of spatial differences in input kinetics on directional tuning, we extended the multicompartmental SAC model to include BCs, modeled as point neurons with center-surround RFs, each proving a single excitatory input to the postsynaptic SAC. Following the space-time model, we considered two distinct populations of BCs innervating the proximal and distal postsynaptic regions (*Figure 2A and B*). BCs within the same population shared identical RF characteristics, but the timing of their responses varied to account for the spatiotemporal progression of the visual stimulus over the simulated circuit.

To investigate which bipolar dynamics promote postsynaptic DS, we employed a machine-learning approach based on evolutionary algorithms (EA, *Figure 2C*; *Ezra-Tsur et al., 2021*). This methodology involved a population of models competing to achieve the training objective. In the initial generation or iteration of the algorithm, the parameters describing RF properties of the proximal and distal BC populations were randomly selected. We stimulated the network with horizontally moving bars and recorded calcium signals on terminal SAC dendrites near the horizontal axis. Direction selectivity index (DSI) was computed from the peak calcium values (*Figure 2B and C*). Initially, DSI levels were very low (mean [± SD] DSI = 7 ± 3%, *Figure 2—figure supplement 1*), with variations among the models due to random initialization. The next generation of candidate solutions was formed by selecting the highest-scoring models to replace the below-average performers. To avoid overfitting, the models were trained with a range of stimulus velocities. Next, the RF properties of the BC population, as well as the global postsynaptic properties (axial resistance, leak, and voltage-gated calcium channel conductance), were subject to random mutations, and this cycle was repeated for 100 generations.

*Figure 2D* illustrates an exemplary solution achieved by the EA, which displays transient distal inputs and slower, delayed proximal BCs. Whole-cell integration of these inputs in the SAC resulted in a significant directional preference (DSI across all velocities was 37 ± 3%, *Figure 2E*).

Notably, the directional tuning profile was found to be dependent on the specified training goal of the model. In our standard configuration, we considered the mean DSI and the amplitude of calcium levels across the recorded dendrites as performance metrics. Including calcium amplitude was crucial in ensuring the robustness of the signal. Models solely rewarded based on DSI levels often evolved strategies that aimed to minimize voltage and calcium levels. This effect was most evident at slow stimulation speeds (*Figure 2—figure supplement 2*). These solutions maximized DSI by capitalizing on the steep nonlinear voltage dependence of the voltage-gated calcium channels near their activation threshold. At these membrane potentials, calcium responses to inward moving stimuli can be almost zero, such that even minute calcium influx at the outward direction will be associated with a significant DSI (*Figure 2—figure supplement 2*). However, the solution is unstable as the DSI is exceedingly sensitive to minor perturbations in local voltage. Consequently, while threshold-based mechanisms can be highly effective, they are also notably susceptible to noise (*Tukker et al., 2004*; *Wu et al., 2023*).

In contrast to the predictions of the threshold-based strategy, physiological somatic excitatory currents (*Figure 2—figure supplement 2*) and dendritic calcium responses in SACs exhibit consistent peak levels across the velocity spectrum (*Ding et al., 2016*; *Koren et al., 2017*). Furthermore, the performance of the threshold-based approach, as measured by DSI, only marginally surpassed the standard configuration (*Figure 2—figure supplement 2*).

## Presynaptic RF parameters influencing optimal directional performance in SACs

To assess the relative contribution of BC kinetics to the resulting DS, we modified the model such that all presynaptic cells shared the same RF formulation. It is worth noting that even with the requirement of identical properties in the bipolar population, the algorithm could explore a vast parameter space by adjusting both the RF properties of presynaptic cells and the postsynaptic integration parameters. However, despite this flexibility, the resulting simulations achieved only a fraction of the DS observed with spatially varying RFs (*Figure 2E*). This result, obtained by integrating realistic BC responses with morphologically detailed SACs, suggests that the space-time wiring model could have a major influence on directional computations in biological circuits.

Using a similar approach, we further investigated the contribution of individual RF parameters to SAC DS. In the modified model described above, all RF parameters in the proximal and distal presynaptic cells were clamped to the same values. Below, we constructed similarly structured models but

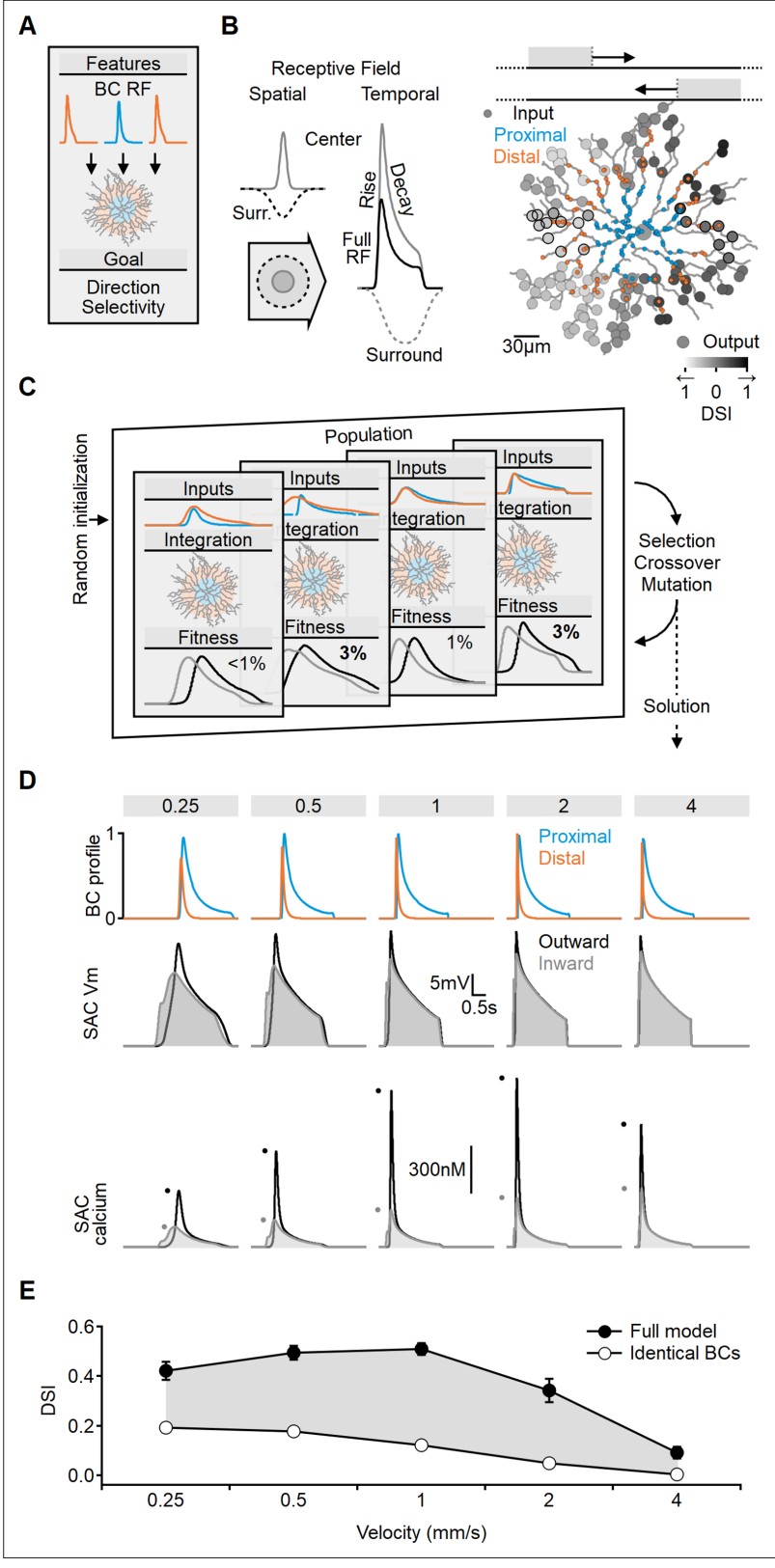

**Figure 2.** Evolutionary algorithm-based model enhances directional voltage responses in starburst amacrine cells (SACs), reproducing key features of the space-time model. (**A, B**) Schematic representation of the bipolar–SAC circuit model. (**B**, Left) demonstrates the spatial components of the bipolar RF center (gray) and surround (dashed) components. The spatiotemporal receptive field (RF) components were convolved with horizontally moving bar

*Figure 2 continued on next page*

*Figure 2 continued*

stimuli (**B**, center) to generate the inputs for the multicompartmental SAC model (**B**, right). Two distinct bipolar groups, each with a unique RF formulation, innervated the proximal and more distal SAC dendrites. (**B**, right) Simulated SAC outputs are color-coded by their direction selectivity index (DSI) levels. The degree of postsynaptic direction selectivity was measured within 30 μm from the horizontal axis (these outputs are highlighted with black strokes). (**C**) The evolutionary algorithm training process involved iterative selection and mutation steps. Each generation included candidate solutions for bipolar RF templates (top row) that were integrated into SAC dendrites (middle row) and ranked based on the directionality and amplitude of calcium signals (bottom row). The best solutions underwent mutation and were propagated to the next generation. (**D**) Example response dynamics of the proximal (blue) and distal (orange) bipolar cells (BCs) (top), representative voltage (middle), and calcium (bottom) signals recorded from a SAC dendrite (location as in *Figure 1*). Dots represent peak response amplitudes in inward (gray) and outward (black) stimulation directions. The model was trained on five velocities (top, units: mm/s). (**E**) Mean (± SD) directional tuning achieved by the model (solid circles, n = 15). Open circles represent the optimal DSI in a bipolar–SAC model with an identical formulation of proximal and distal BCs. In this scenario, direction selectivity is mediated by voltage filtering in SAC dendrites.

The online version of this article includes the following figure supplement(s) for figure 2:

**Figure supplement 1.** Example evolution of the directional tuning in a bipolar–starburst amacrine cell (bipolar–SAC)model.

**Figure supplement 2.** Limitations of model training on direction selectivity index (DSI) alone.

allowed a single parameter to differ between the BC populations. Previous studies demonstrated that space-time wiring yields the highest DS levels when presynaptic units exhibit different activation lags or response transiency (*Fransen and Borghuis, 2017*; *Kim et al., 2014*; *Matsumoto et al., 2019*; *Srivastava et al., 2022*). Consistent with this, models in which proximal and distal BCs differed only in their response delay or decay times showed DSI levels close to those observed in the full model (29 ± 2% and 35 ± 3%, respectively, *Figure 3*).

Notably, temporal differences in BC responses were not solely influenced by these parameters; we also found that the spatial extent of the RF could introduce an activation lag, which was inversely proportional to RF size: neurons with wider RFs can respond to a moving object sooner compared to their immediate neighbors with smaller RFs. In line with this observation, the optimal solutions in the full model evolved toward larger RF centers in the distal BCs (42 ± 15 μm vs. 32 ± 9 μm, *Figure 3C*). The difference in RF size was more pronounced when center width was the only varying RF parameter (97 ± 18 μm vs. 22 ± 3 μm), resulting in a measurable lag (approximately 135 ms for stimulus velocity = 0.5 mm/s, *Figure 3*). The impact of spatial RF properties on directional tuning was not as pronounced as the temporal RF characteristics but managed to elevate the DS by about twofold over the space-invariant configuration (DSI = 19 ± 4%, p<10$^{-11}$ vs. identical BCs, *Figure 3C*).

## The space-time wiring model does not require dendritic isolation

Previous studies investigating the postsynaptic mechanisms underlying DS in SACs have highlighted the importance of electrotonic isolation of terminal dendrites in promoting directional signals (*Ding et al., 2016*; *Koren et al., 2017*; *Poleg-Polsky et al., 2018*). However, since the space-time wiring mechanism primarily operates on presynaptic circuits, it may be less reliant on postsynaptic compartmentalization. To test this hypothesis, we manipulated inter-dendritic signal propagation by modifying the axial and membrane resistance properties of the model in segments around the soma (*Figure 4*).

As anticipated, increasing the axial resistance led to decreased somatic depolarization during simulated visual responses (*Figure 4A*). The disparity in voltage attenuation between the control configuration and the model with elevated axial resistance was particularly pronounced when stimulating a single SAC branch, confirming the near-perfect isolation of dendrites (*Figure 4B*). Conversely, reducing the barrier to signal propagation had the opposite effect (*Figure 4A and B*). Despite these variations in perisomatic electrical properties, all models generated by the EA achieved comparable DS levels (*Figure 4C*).

Collectively, these findings suggest that the effectiveness of the space-time wiring model relies primarily on BC kinetics, which are influenced by the spatiotemporal properties of their RF organization. In contrast, the specific details of postsynaptic integration appear to play a lesser role in this DS mechanism.

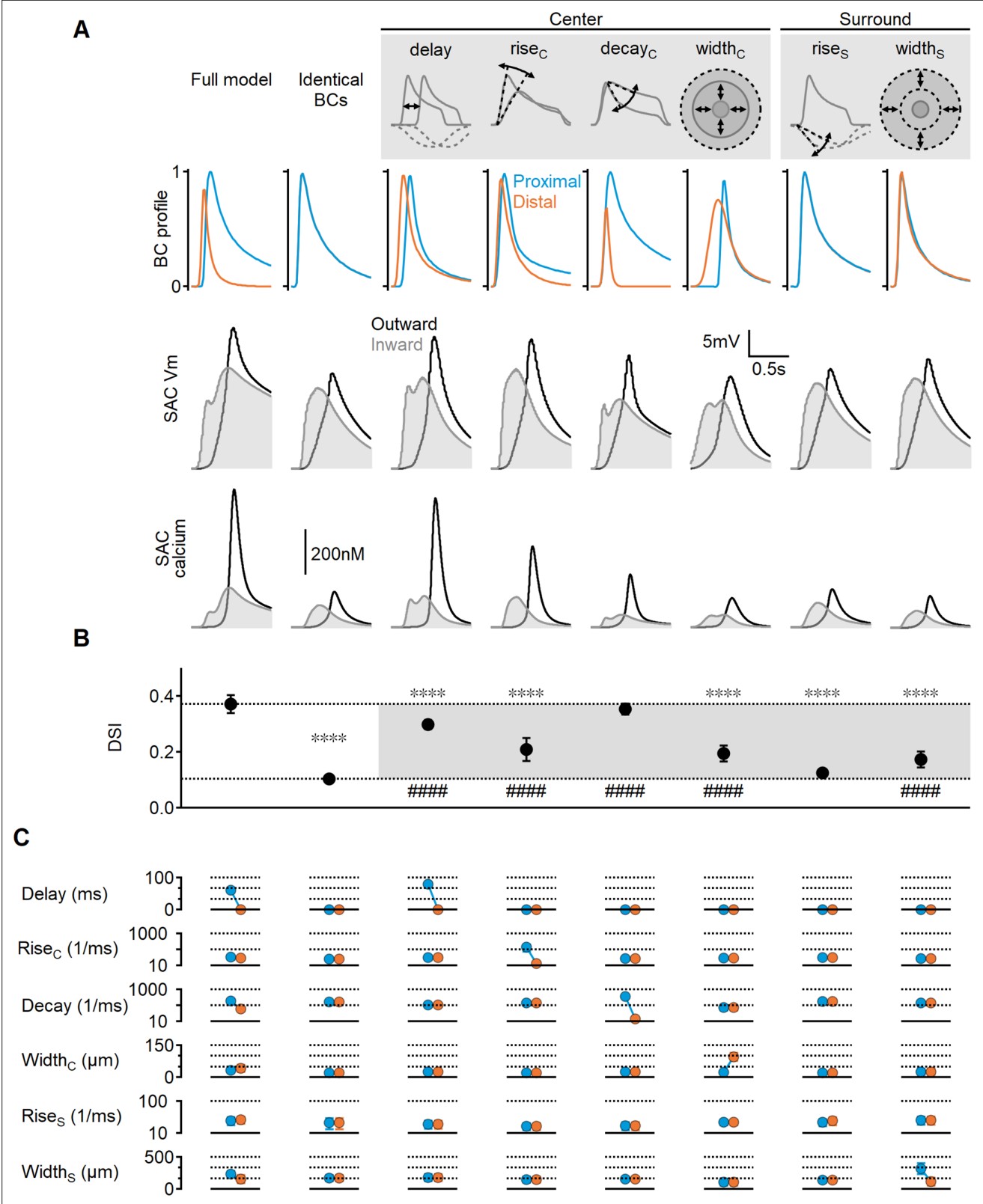

**Figure 3.** Impact of bipolar cell (BC) receptive field (RF) components on direction selectivity (DS) performance. (**A**) Representative responses of bipolar dynamics (top), voltage (middle), and calcium (bottom) in starburst amacrine cell (SAC) dendrites (stimulus speed = 0.5 mm/s) in a model where the proximal and distal bipolar RF formulations differed in a single parameter: response lag ('delay'), rise/decay kinetics, or the spatial extent of RF components. The original, unconstrained configuration and a model with identical RFs are included for comparison. (**B, C**) Mean (± SD) values of

*Figure 3 continued on next page*

*Figure 3 continued*

direction selectivity index (**B**) and the distribution of RF parameters in the proximal/distal presynaptic groups (**C**) for each scenario (n = 10). Dotted lines in (**B**) indicate the mean values of the full and identical RFs modes. Rise/decay kinetics are presented on a logarithmic scale. The spatial extent of the center and surround RF components is expressed as the full width at half maximum (FWHM). ∗∗∗∗p<10$^{-6}$ vs. the full model. ####p<10$^{-6}$ vs. the identical RF model (ANOVA followed by Tukey's test).

## Diversity of glutamatergic inputs in murine SACs

In the previous sections, we examined the theoretical implementation of the space-time wiring hypothesis through detailed simulations of the bipolar–SAC circuit. To investigate the similarity between the RF properties of mouse BCs innervating SACs and the optimal RFs predicted by our models, we conducted glutamate imaging using a two-photon microscope on an ex vivo retina preparation. To specifically target ON- and OFF-SACs, we virally expressed floxed iGluSnFR in ChAT-Cre mice (*Figure 5*). Then, 3–6 wk after vector introduction, we performed ex vivo experiments using a whole-mount retina preparation. Visual stimuli consisted of full-field flashes and bars moving alternatively either from left to right or in the opposite direction at five different velocities (*Figure 5B*) and oriented bars used to map RFs using the filtered back-projection approach (*Figure 5C*).

To analyze the diversity of BC signaling in response to motion, similarly responding pixels were grouped into regions of interest (ROIs) (*Figure 5A and B*). The time of response onset depends on intrinsic RF properties shared among BC subtypes and the position of the RF relative to the moving bar stimulus. To disentangle these components, we aligned the responses based on the mean 50% rise time observed during stimulation in both leftward and rightward directions. This alignment negated the dependency on the relative position of the RF, allowing us to focus solely on differences in response kinetics to motion.

Next, we analyzed common response motifs across recording regions. We identified functional release clusters using hierarchical clustering of the aligned and normalized dF/F waveforms evoked by a single motion velocity (0.5 mm/s). This procedure was performed separately for ON and OFF ROIs (n = 334 and 135 regions, respectively, *Figure 6A and C*). We determined the optimal number of functional clusters from the curve describing the within-cluster variance (*Figure 6—figure supplement 1*). The analysis identified seven groups for the ON-SAC population (*Figure 6A*, *Figure 6—figure supplement 1*) and six for the OFF-SACs (*Figure 6C*, *Figure 6—figure supplement 1*) as providing the optimal separation (*Franke et al., 2017*; *Gaynes et al., 2022*; *Matsumoto et al., 2021*; *Matsumoto et al., 2019*; *Rasmussen et al., 2020*; *Strauss et al., 2022*). As BC dynamics vary systematically with axonal stratification level in the IPL (*Franke et al., 2017*), we sorted the functional clusters based on their transiency of the response dynamics, with C1 representing the most transient shape (*Figure 6B and D*).

## Mismatch between response lags measured from flashed and moving stimuli

The original 'space-time' model identified response delay, also known as 'lag', as the primary factor influencing directional tuning (*Kim et al., 2014*). The interval between the presentation of visual stimuli and the neural response varies systematically among distinct BC subtypes (*Baden et al., 2013*). *Kim et al., 2014*, proposed that Hassenstein–Reichardt correlator is implemented in the bipolar–SAC circuit with a combination of delayed proximal and short-lag distal BCs. Lags are typically identified from flash responses. In order to contribute to motion computations, the lag should be an immutable feature of the visual response and persist with moving stimuli. Given the difference in signal transformation between motion and flash responses (*Figure 7—figure supplements 1 and 2*; *Gaynes et al., 2022*), we sought to investigate whether the onset of glutamate response also depends on stimulus characteristics. In *Figure 7A*, we present the correlation between response delay, measured within the same ROIs for all glutamate clusters, for both moving and stationary stimuli. For flashes, the lag was determined as the interval between the appearance of the stimulus and the initiation of the fluorescent response. In the case of moving stimuli, we measured the interval starting from the time the stimulus swept over the position of the cell.

Unexpectedly, our experimental findings have uncovered a reversed relationship in the response delays (Pearson correlation coefficient, *r* = –0.98). Functional clusters with the longest lags in response to flashes tended to respond quickest to moving bars (*Figure 7A*). In certain clusters, we observed

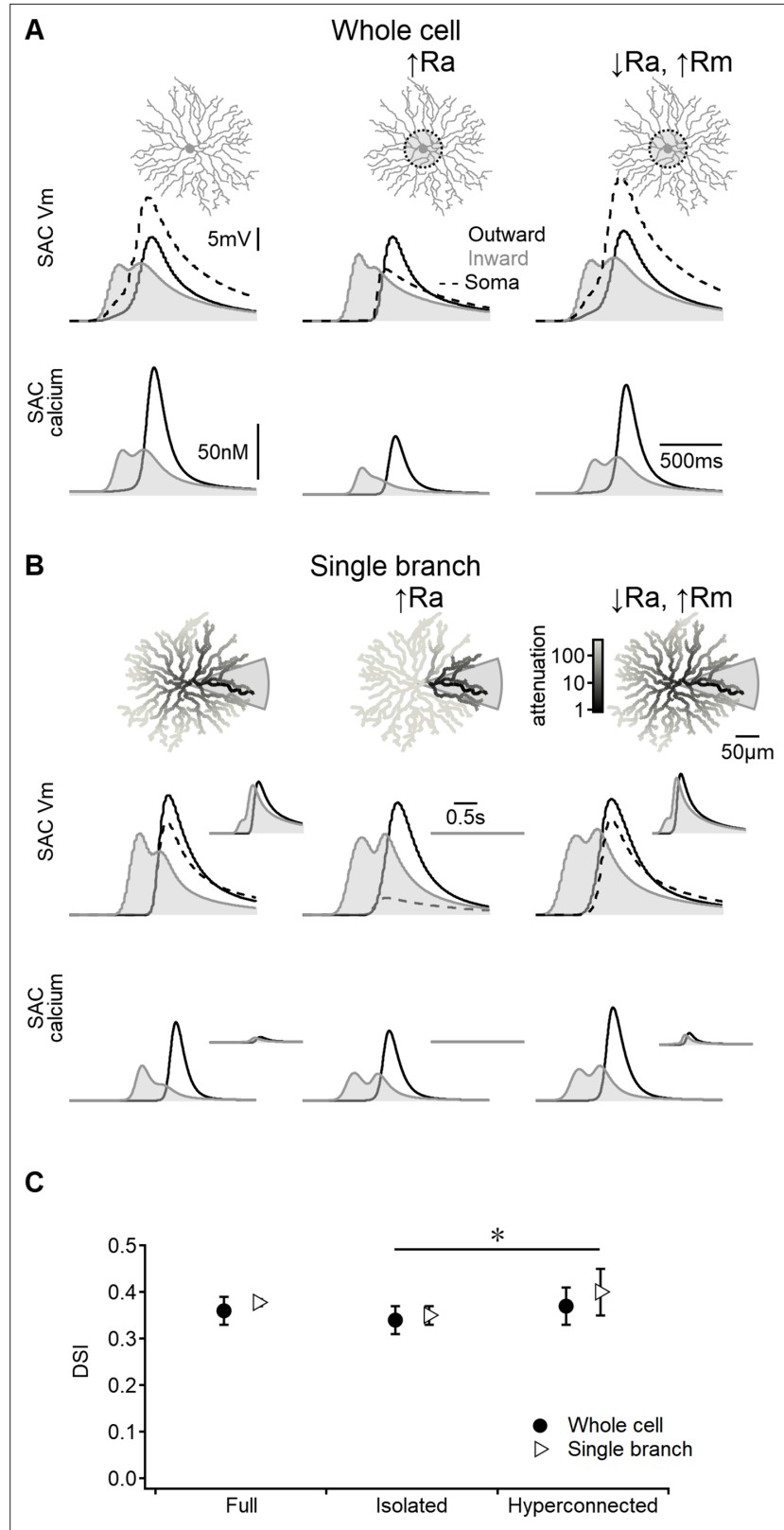

**Figure 4.** Direction selectivity (DS) in the space-time wiring model is independent of dendritic isolation. (**A**) Representative voltage profiles and dendritic calcium signals in the original model (left), starburst amacrine cell (SAC) with reduced inter-dendritic interactions due to elevated internal resistance in the perisomatic area (middle), and a 'hyperconnected' SAC model with low signal attenuation in the dendritic tree (right). (**B**) Similar to panel

*Figure 4 continued on next page*

*Figure 4 continued*

(**A**) but for models evolved to enhance DS signals in a single stimulated branch. The cell morphology is color-coded based on voltage attenuation from a distal release site of the stimulated dendrite. Insets show voltage and calcium signals recorded on the opposite side of the cell. (**C**) Summary of directional tuning observed with different levels of compartmentalization, suggesting a minimal impact of isolation on SAC performance. *p=0.01 (ANOVA followed by Tukey's test).

negative delays in response to motion, indicating that glutamate release begins before the moving bar reaches the position of the cell (*Berry et al., 1999*; *Trenholm et al., 2013*). Negative delays are more likely in cells with wide RFs, leading us to suspect that early responders to motion possess spatially extensive RFs. Our observations supported this hypothesis as we discovered a clear relationship between the lag in response to moving stimuli and the measured RF half-width ($r = -0.99$, *Figure 7B*).

Taken together, our results indicate a significant difference between the activation lag following a flashed and moving stimuli. The spatial properties of the RF play a significant role in determining the

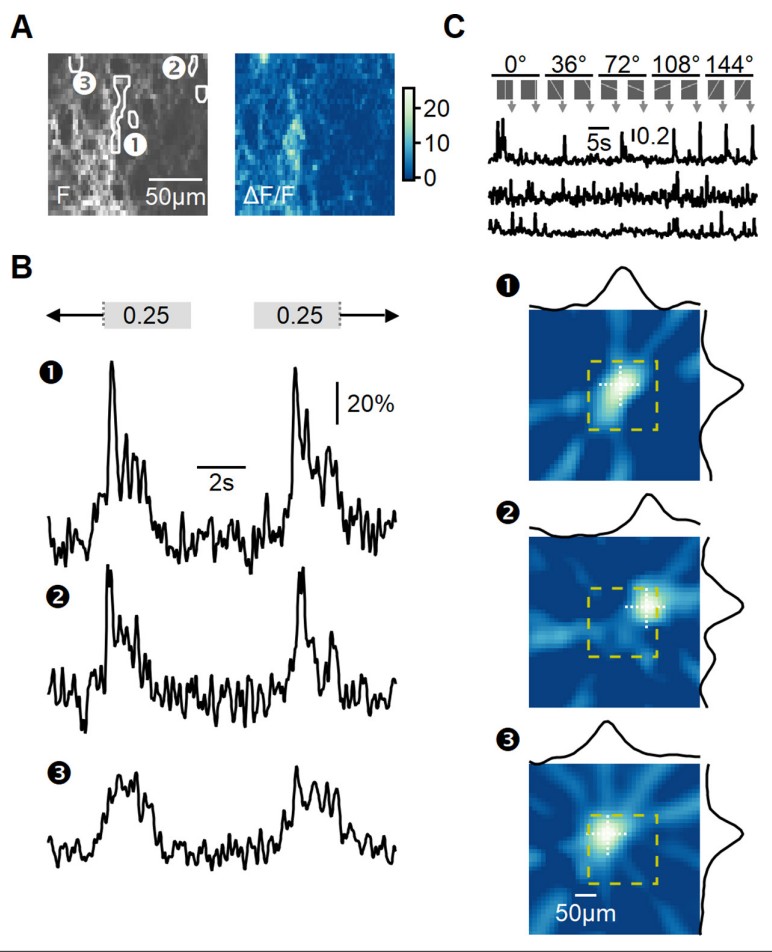

**Figure 5.** Recording of glutamatergic drive to starburst amacrine cells (SACs) during full-field motion. (**A**) Two-photon image of one example field of view (FOV) displaying the average iGluSnFR fluorescence (left) and the processed dF/F signals (right). Floxed iGluSnFR expression was induced using AAVs in ChAT-Cre mice. (**B**) Responses from the regions of interest (ROIs) indicated in panel (**A**) to horizontally moving bars (speed = 0.25 mm/s). (**C**) Receptive field (RF) mapping using the filtered back-projection technique. Top: changes in fluorescence from three example ROIs in response to bars flashed at 32 different spatial positions and 5 orientations. Bottom: reconstructed spatial RFs. The yellow square represents the estimated extent of the two-photon FOV shown in panel (**A**). The black curves represent the x and y RF profiles measured at the center of mass (indicated by white dotted lines).

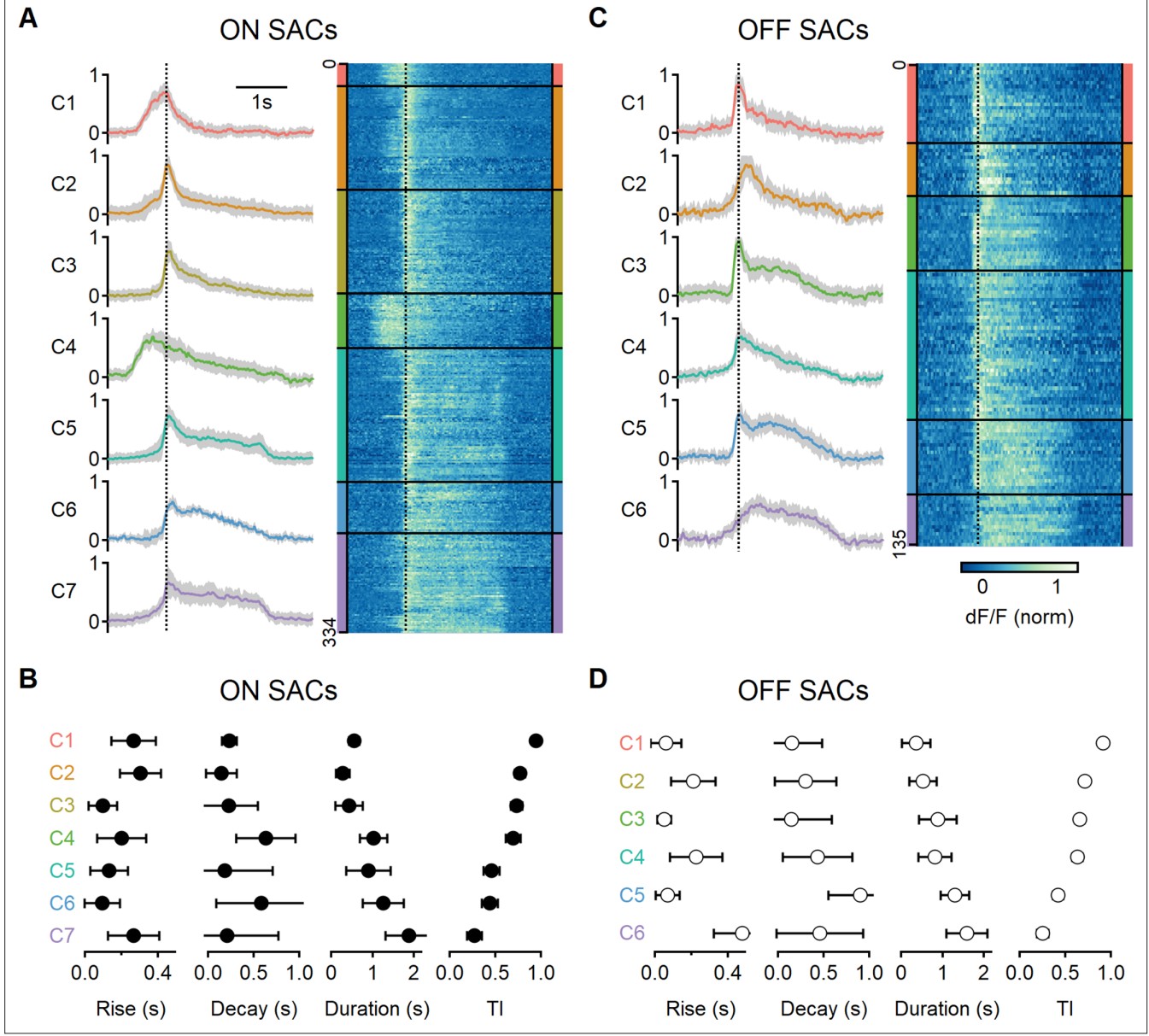

**Figure 6.** Diversity of glutamatergic responses to motion in ON- and OFF-SAC populations. (**A, C**) Left: the average glutamate signals in functional clusters determined from regions of interest (ROI) responses to motion (speed = 0.5 mm/s, color-coded by cluster identity). Shaded areas mark the standard deviation. The dotted line indicates the time of peak response of cluster C1. Right: heatmaps of the responses from individual ROIs. (**B, D**) Mean (± SD) waveform characteristics measured from individual ROIs in each cluster. TI, transiency index. Clusters are sorted based on their transiency. SAC, starburst amacrine cells.

The online version of this article includes the following figure supplement(s) for figure 6:

**Figure supplement 1.** Determining the optimal number of functional clusters for the ON- and OFF-SAC populations.

**Figure supplement 2.** Stratification profile of the functional clusters detected from motion responses in ON- and OFF-SACs.

**Figure supplement 3.** Comparison of response onset from different functional clusters.

delay for motion responses but have no impact on the lag seen following the presentation of full-field flashes, which engage the entire RF regardless of its size.

## Determination of RF characteristics from motion responses

To gain deeper insights into the signal processing underlying the glutamatergic signals to SACs, we set to study the RF composition of the functional clusters. Toward this task, we decided to train

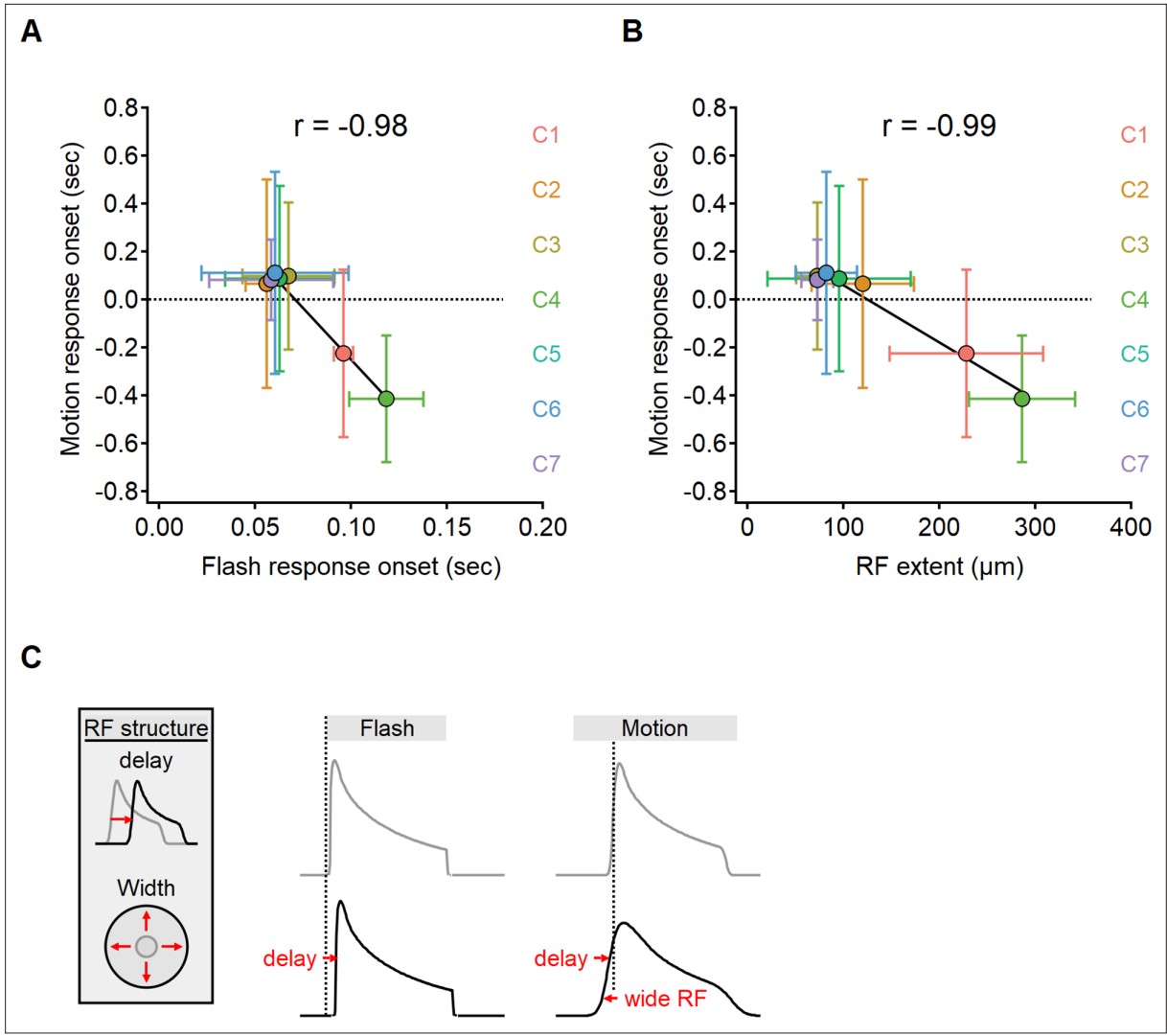

**Figure 7.** The onset of motion responses depends on the extent of the receptive field (RF) and not on response lag measured with static stimuli. (**A, B**) Correlation between the onset of motion responses and the static response lag (**A**), and RF width (**B**) color-coded by functional cluster identity. Clusters with the longest lags have wide RFs and earliest responses to motion. (**C**) Illustration of the interaction between two mechanisms contributing to the time of response onset. Cells with prolonged visual processing delay will respond later to the presentation of a static stimulus (black). When moving bars are presented, response timing depends on the processing time lag, the size of the RF, and stimulus velocity.

The online version of this article includes the following figure supplement(s) for figure 7:

**Figure supplement 1.** Comparison of response dynamics to moving bars and stationary flashes.

**Figure supplement 2.** Example receptive field (RF) shape and responses to moving and static stimuli with full-field and masked stimulation.

center-surround RF models to match the shapes of the recorded waveforms. We adapted model formulation described above for simulating BC dynamics, with the addition of a filter to account for iGluSnFR binding and unbinding dynamics, thereby simulating the sensor-mediated filtering of glutamate signals drive (*Armbruster et al., 2020*; *Hain and Moser, 2023*; *Srivastava et al., 2022*). Separate EA models were trained for each functional cluster, utilizing five stimuli with different velocities to drive the model. The objective of the training was to replicate the shape of the recorded fluorescent signals during the presentation of moving bars at corresponding velocities (*Figure 8A*).

Using this approach, we were able to faithfully reproduce the empirical glutamate dynamics observed in the training dataset (*Figure 8A*, *Figure 8—figure supplement 1*). As an initial validation of the algorithm's predictive power, we compared glutamate signals to the response of the RF models measured during the presentation of full-field flashes (*Figure 8B*). Encouragingly, even though these stimuli were not part of the training set, the simulated output reliably replicated experimental

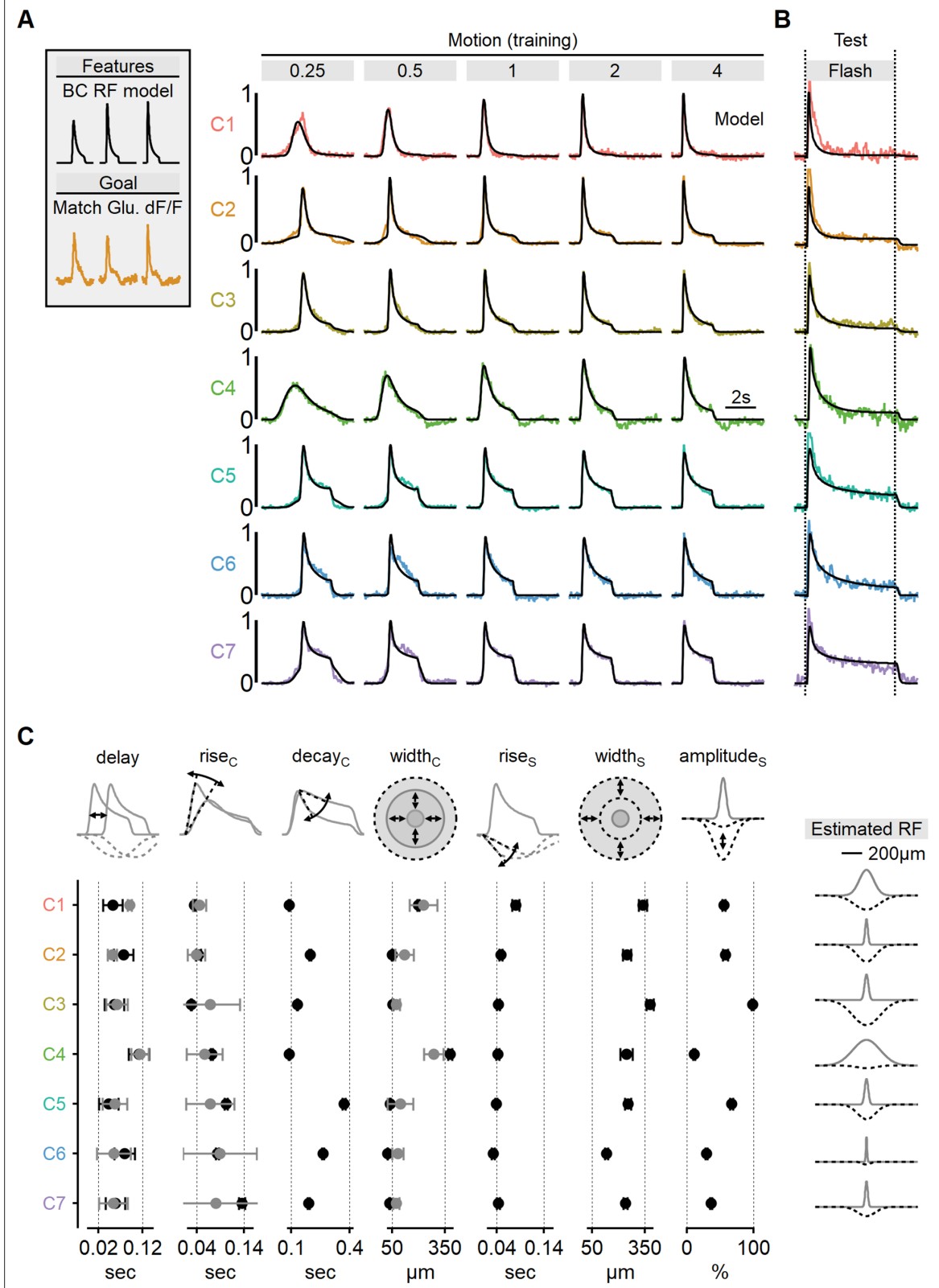

**Figure 8.** Estimated receptive field (RF) properties from presynaptic responses to motion. (**A**) Center-surround RF model was trained using an evolutionary algorithm to match experimentally recorded waveforms in ON-SACs. Experimental data is color-coded as in *Figure 6*. The output of the seven models is shown in black. (**B**) Comparison between experimentally recorded responses to 4-s-long full-field flashes and the predictions of the models (black). (**C**) Mean (± SD) RF properties measured from each of the models (n = 10 repeats for each cluster) are shown in black. The

*Figure 8 continued on next page*

*Figure 8 continued*

corresponding parameters determined experimentally are shown in gray. Delay and rise time were measured from flash responses, and the full width at half maximum (FWHM) of the center was analyzed from RF maps. The right panel illustrates the predicted spatial extent of the center (gray) and surround (dotted black) RF components in each functional cluster. SACs, starburst amacrine cells.

The online version of this article includes the following figure supplement(s) for figure 8:

**Figure supplement 1.** Estimated receptive field (RF) properties from presynaptic responses to motion in OFF-SACs.

waveforms, despite substantially different dynamics in many clusters (*Figure 8B*, *Figure 8—figure supplement 1*).

By leveraging the more interpretable stimulus-response dynamics of the full-field flash paradigm, we could examine the match in individual model parameters to experimental responses. We found a strong agreement between the rise time of the flash response and the predicted kinetics of RF centers across all functional clusters (*Figure 8C*). Similarly, response delay in the models reliably followed the lags seen in flash responses (*Figure 8C*).

As a final validation of the novel RF mapping approach we developed, we examined the spatial characteristics of the glutamate signals. Our models consistently converged on narrow RFs, typically ranging from 50 to 100 μm for the majority of functional clusters, consistent with the classical RF description for retinal BCs (*Figure 8C*, *Figure 8—figure supplement 1*; *Euler et al., 2014*; *Franke et al., 2017*; *Kuo et al., 2016*; *Schwartz et al., 2012*; *Strauss et al., 2022*; *Turner et al., 2018*; *Wienbar and Schwartz, 2018*). Finally, in a close match with experimentally mapped RFs (*Figure 7B*), the models predicted significantly more extensive RFs for ON-C1 and ON-C4, with half-widths of 379 ± 13 and 198 ± 6.7 μm, respectively (*Figure 8C*), strongly suggesting that RF formulation derived from motion responses is accurate and could be used to predict signal processing in untrained visual conditions.

## Realistic BC dynamics suggest a modest effect of the space-time wiring on directional tuning in ON- and OFF-SACs

After establishing the diversity of excitatory dynamics during visual motion, we proceeded to examine the extent to which their combination could enhance DS in SACs. To achieve this, we replaced the synthetic bipolar description in the bipolar–SAC EA model with fits to experimentally determined glutamatergic waveforms. These fits provided a deconvolved representation of the excitatory drive, which is more likely to approximate the actual presynaptic signals reaching the SACs. We note that the difference between the recorded and deconvolved waveforms was minimal, and utilizing the experimentally recorded shapes did not significantly impact the subsequent results (data not shown).

We first considered a simplified case where two distinct bipolar populations targeted the proximal and more distal postsynaptic regions. The timing of bipolar responses varied to account for the spatiotemporal progression of the visual stimulus over the simulated circuit. *Figure 9* illustrates the peak DS index achieved with EAs trained to find the postsynaptic properties (axial resistance, leak, and voltage-gated calcium channel conductance) that lead to optimal directional performance for all possible proximal and distal functional cluster placement combinations. In line with prior work (*Fransen and Borghuis, 2017*; *Kim et al., 2014*; *Srivastava et al., 2022*; *Stincic et al., 2016*; *Wu et al., 2023*), the optimal DS was attained when proximal BCs with sustained waveforms were combined with transient distal cells. For ON-SACs, the highest DSI = 17 ± 2% was observed with inputs from ON-C6 (proximal) and ON-C2 (distal). In the case of OFF-SACs, the inputs performed slightly better, with the highest DSI levels (21 ± 3%) associated with OFF-C1 and OFF-C5 (*Figure 9*). In contrast, SACs innervated by BCs with identical RF formulation had substantially lower directional capabilities (DSI = 5 ± 1% for ON-SACs and 7 ± 1% for OFF-SACs, *Figure 9*). As in our previous experiments, we permitted the models to adjust both passive and active postsynaptic properties. Our observations yielded a result consistent with what we observed with synthetic BC inputs: the degree of measured DS did not exhibit a significant correlation with the extent of dendritic isolation (*Figure 9—figure supplement 1*).

It is possible that the restriction to innervation by two input populations precludes the model from converging on a better solution. To take full advantage of the heterogeneity of the experimentally measured signals, we examined if directional outcomes improve when SACs are allowed to integrate over more presynaptic clusters. Inspired by a previous model (*Fransen and Borghuis,*

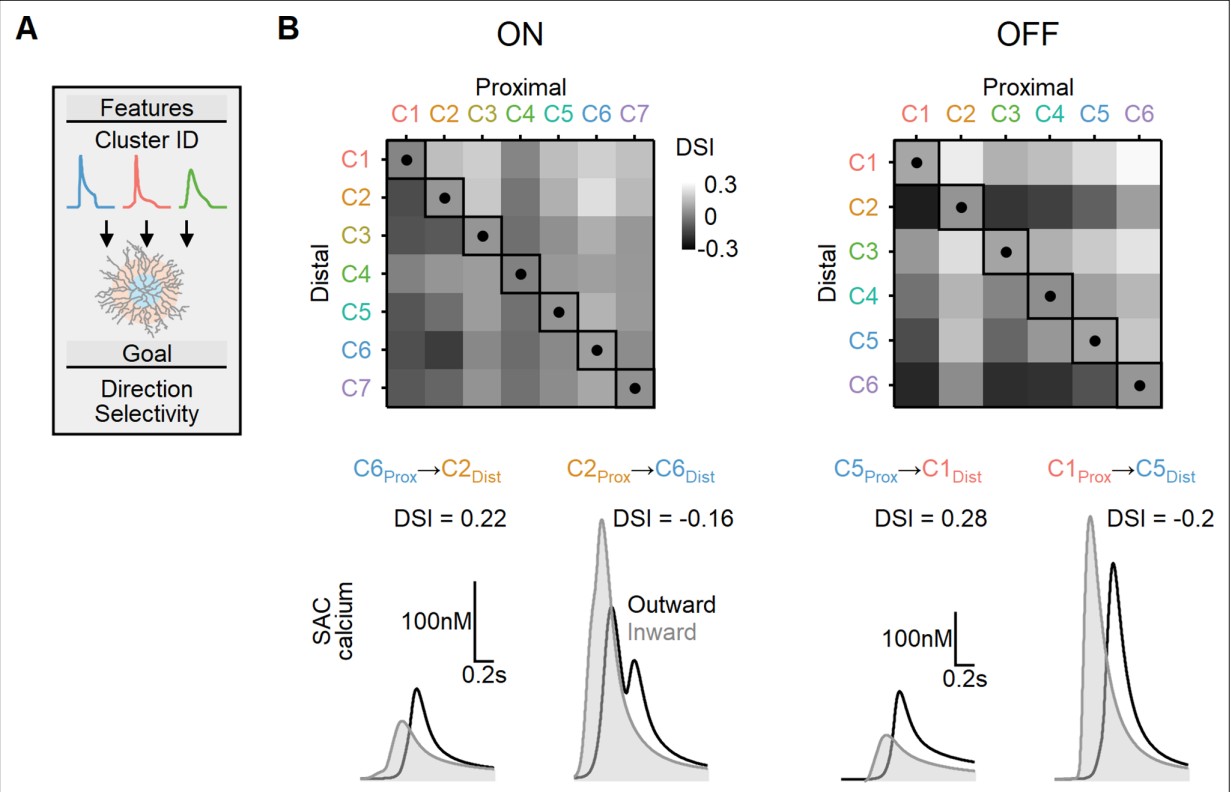

**Figure 9.** Directional tuning in bipolar–starburst amacrine cell (SAC) models with experimentally recorded excitatory waveforms segregated into proximal-distal regions. (**A**) Schematic representation of the evolutionary algorithm employed to maximize direction selectivity (DS) by utilizing deconvolved waveforms from experimentally recorded clusters as the input. (**B**) Top: DS achieved by the models when various combinations of input waveforms are targeted toward the proximal and distal SAC dendrites. Squares and dots represent cases where the waveforms are identical for all bipolar cells. Bottom: representative calcium signals obtained from the best (left) and worst (right) combinations of input waveforms.

The online version of this article includes the following figure supplement(s) for figure 9:

**Figure supplement 1.** Correlation between directional tuning and postsynaptic passive and active parameters.

*2017*), we distributed presynaptic BCs over concentric annuli, preserving input symmetry along the soma-dendritic tip axis (*Figure 10A*). Specifically, we grouped the inputs into 10-µm-wide bins based on their distance from the postsynaptic soma. Each bin was assigned a single functional cluster RF description drawn at random. Now, the search algorithm evolved to determine the spatial distribution of the recorded functional clusters that produced the most robust directional tuning across a range of stimulus velocities (*Figure 10*).

We observed that the simulations converged on similar solutions, irrespective of the initial randomized cluster distribution. Optimal DS was achieved in both ON and OFF circuits when two clusters dominated the proximal and distal innervation regions (*Figure 10B*). Consequently, the levels of DS obtained in these simulations did not significantly differ from the findings of the simple proximal-distal parameter exploration presented earlier (ON-SACs: DSI = 17 ± 5%, OFF-SACs: DSI = 21 ± 6%, *Figure 10C–F*). Based on these results, we can conclude that optimal combination of excitatory signals has the potential to triple the level of directional tuning over SAC-intrinsic mechanisms (*Figure 10D and F*; *Tukker et al., 2004*; *Vlasits et al., 2016*; *Wu et al., 2023*).

Why do the solutions found with experimentally recorded glutamate waveforms underperform compared to the synthetic model? Focusing on the clusters identified by our analysis as contributing to the highest directional tuning, we note that although distal clusters (ON-C2 and OFF-C1) had the shortest duration and fastest delays, their kinetics and RF description were about twofold longer than the optimal synthetic distal inputs (*Figure 11*, *Figure 11—figure supplement 1*), with the largest difference being the rise time dynamics of ON-C2 (303 ± 110 ms vs. 45 ± 12 ms for optimal model). Even further pronounced were the differences in recorded vs. optimal proximal solutions. The

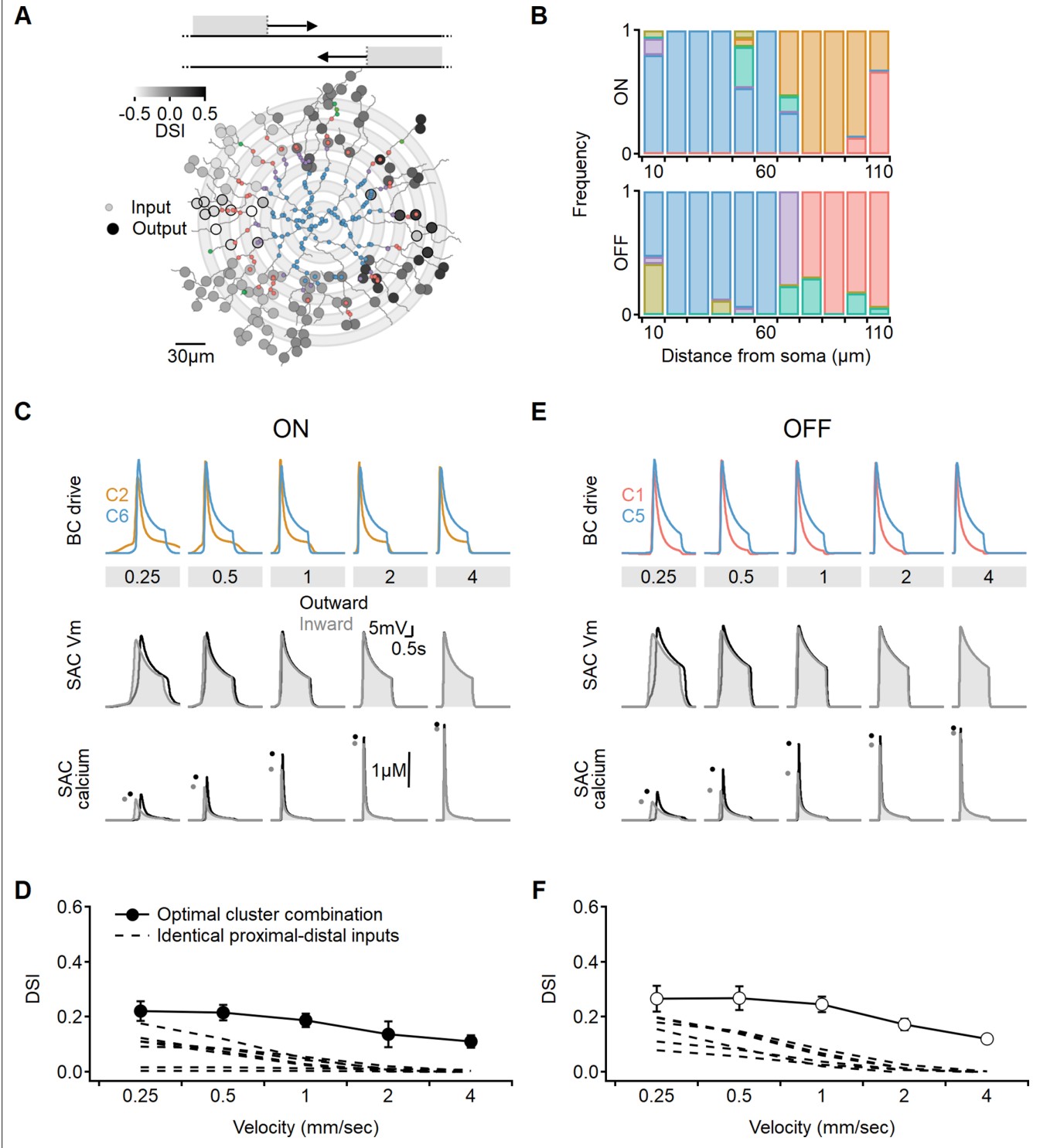

**Figure 10.** Optimal direction selectivity (DS) with experimentally recorded excitatory waveforms. Investigation of directional performance in a multicompartmental bipolar-SAC model innervated by experimentally-recorde functional clusters. (**A**) Illustration of an example solution superimposed on SAC morphology. The distribution of BC inputs was symmetrical along the soma-dendritic axis based on their distance from the soma (small circles and grey annuli). Deconvolved waveforms from one of the experimentally recorded clusters were applied to all synapses within each bin. Output synapses are indicated with large circles and color coded by the DSI. The degree of postsynaptic direction selectivity was measured within 30 µm from the horizontal axis (these outputs are highlighted with black strokes). (**B**) The spatial distribution of functional clusters producing optimal directional signals across 15 model runs. Input color coding in (**A**) and (**B**) as in *Figure 6*. (**C**) Top: overlay of the velocity tuning dynamics for ON-C2 and ON-C6, representing the most commonly observed proximal and distal clusters. Middle and bottom: voltage and calcium signals in a SAC dendrite generated

*Figure 10 continued on next page*

by one of the evolved models. Dots indicate peak response amplitudes in inward (grey) and outward (black) stimulation directions. (**D**) Directional tuning as a function of stimulus velocity, with the solid curve representing the mean (± SD) results obtained from the model. Dotted, velocity tuning calculated from simulations with SAC innervation by a single functional cluster (see *Figure 9*). 'Space-time wiring' improves directional selectivity over a wide range of stimulation velocities. (**E-F**) as in (**C-D**), but for OFF-SACs.

durations of ON-C6 and OFF-C5 were 1.25 ± 0.49 s/1.28 ± 0.34, respectively, compared to 0.31 ± 0.15 s measured for synthetic RFs. Similarly, the transiency indexes calculated for these clusters were around 0.4, while optimal synthetic waveforms had TI = 0.87 ± 0.09 (*Figure 11*). Thus, glutamate release in the bipolar–SAC circuit has slow dynamics that can mediate substantial, albeit suboptimal, DS.

# Discussion

In this study, through a combination of experimental approaches and computational modeling, we have explored the theory and the evidence for bipolar organization supporting the space-time wiring model. The original space-time wiring hypothesis was conceptualized for individual SAC dendrites, and the difference in BC response shape was considered through the lens of the dynamics of responses to static stimuli (*Kim et al., 2014*). Although these circumstances were thought to simplify the analysis of dendritic computations, it is now clear that signal preprocessing in the bipolar population introduces unforeseen challenges to this approach: our prior work demonstrated a low correlation between the dynamics of BC responses to motion and static stimuli (*Gaynes et al., 2022*). Here, we replicate this result on a new dataset (*Figure 7—figure supplement 2*), reinforcing the notion that signal kinetics, crucial to the space-time wiring model, are not a fixed property of the cell but are instead dependent on the stimulus. For example, we demonstrate that presynaptic signals can exhibit prolonged flash-lags yet generate early responses to motion, a phenomenon explained by their spatial RF structure (*Figure 7*). In addition, it is vital to consider the implications of using stimuli that activate only a small portion of the SAC dendritic tree in isolation or all dendrites through the same stimulation pattern. Such stimulus designs are likely to introduce nonlinear signal processing in BCs and, importantly, result in asymmetric bipolar responses for different stimulation directions (*Gaynes et al., 2022*;

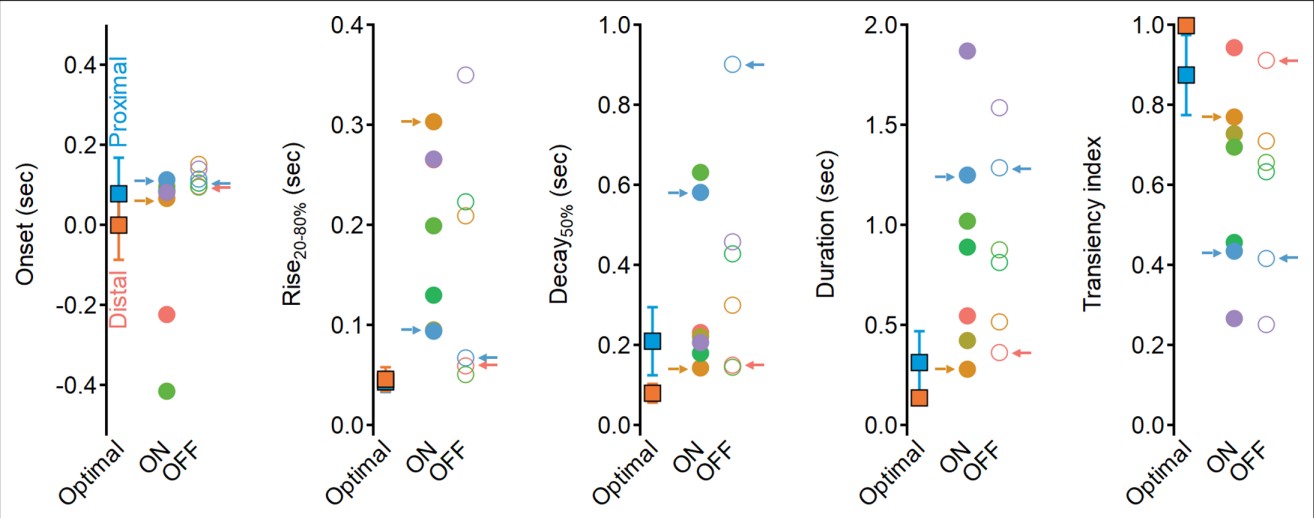

**Figure 11.** Dynamics of glutamate release onto starburst amacrine cells (SACs) are sluggish and more sustained compared to the optimal excitatory drive produced in synthetic model. Waveform parameters, measured for synthetic receptive field (RFs) driving optimal direction selectivity (squares) and for experimentally determined functional clusters (filled circles – ON, open circles – OFF). Color coding as in *Figure 6*. Clusters identified as best contributors for the Hassenstein–Reichardt correlator (proximal: ON-C2 and OFF-C1; distal: ON-C6 and OFF-C5) are marked with arrows.

The online version of this article includes the following figure supplement(s) for figure 11:

**Figure supplement 1.** Comparison of receptive field (RF) structure between experimentally determined clusters and the synthetic model mediating optimal directional sensitivity.

*Strauss et al., 2022*). This poses a concern to the space-time model as it assumes consistent presynaptic waveforms independent of the stimulus direction (*Kim et al., 2014*).

To overcome these issues, it becomes crucial to devise visual stimuli that effectively bypass both limitations. One approach is to design stimuli that either appear or begin to move outside the RF of the BCs innervating the postsynaptic cell. By doing so, we can ensure that the stimulation does not trigger asymmetric responses and maintain the consistency of presynaptic waveforms, as required by the space-time model. A simple stimulus that fulfills this criterion is a continuously moving object that appears and disappears at a considerable distance (several hundred microns) from the cells of interest (*Gaynes et al., 2022*; *Wu et al., 2023*). By employing this stimulus design, we can effectively mitigate the nonlinear signal processing in BCs and preserve the foundational assumption of consistent presynaptic waveforms in the space-time model. This approach allows for more accurate investigations into the dynamics of BC responses across velocities and their contribution to the directional tuning in SACs and DSCGs.

While the use of full-field stimuli restricts the ability to investigate the responses of individual branches with high granularity, it provides an opportunity to explore a broader question: the potential impact of interactions among multiple dendrites on DS. Dendritic compartmentalization in SACs is promoted by morphological factors (*Tukker et al., 2004*; *Wu et al., 2023*) and by active processes, such as the presence of potassium channels (*Ozaita et al., 2004*). Additionally, in the mouse, SAC-SAC inhibition predominantly occurs in the perisomatic region, where it modulates communication between branches and maintains the output synapses of SACs within the optimal range for effective DS processing (*Ding et al., 2016*; *Poleg-Polsky et al., 2018*). Similar enhancement of DS signaling is mediated by metabotropic glutamate receptors. By limiting calcium influx, they effectively reduce the spread of electrical signals within the dendritic arbor, enhancing the electrotonic isolation of individual processing units (*Koren et al., 2017*). Interestingly, our results demonstrate that in contrast to these postsynaptic mechanisms, compartmentalization has little effect on directional tuning produced by the space-time wiring model (*Figure 4*). This unexpected outcome highlights the distinction between pre- and postsynaptic computations within the bipolar–SAC circuit.

In this study, we employed EAs, a powerful machine-learning method, to investigate the mechanisms of DS computations in SACs. EAs have broad applicability and have been successfully utilized in examining the mechanisms of directional computations of SACs before (*Ezra-Tsur et al., 2021*). Our work advances the mechanistic understanding of the contribution of the space-time wiring model to stimuli that engage the dendritic tree in an asymmetric manner, experimental determination of the release of glutamate from presynaptic cells, and the amplification of signals by postsynaptic calcium channels. While EAs place minimal constraints on the model type or examined features, it is important to note that these algorithms are most effective when dealing with models with a relatively constrained feature space. While they may be susceptible to getting trapped in local minima, our largest model, which consisted of 23 free parameters (9 × 2 presynaptic + 5 postsynaptic, as described in 'Methods'), consistently evolved toward similar solutions regardless of the starting parameters. In cases where EAs do not reliably converge on an optimal solution, alternative machine-learning approaches can be employed. For instance, a recent study utilized a neural network system that combined Hassenstein–Reichardt correlators with biologically inspired RFs to describe solutions for computations in collision avoidance neurons of flies (*Zhou et al., 2022*).

We present a novel application of model fitting to map RFs based on responses to moving stimuli. Previous studies have used stimuli moving in different directions to estimate the spatial location of the RF (*Gaynes et al., 2022*; *Wienbar and Schwartz, 2018*). We expand upon this approach by incorporating multiple stimulus velocities. This extension allows efficient RF determination from a brief visual stimulation (approximately 2–3 min). However, it is essential to acknowledge that our fits to the data were based on the assumption of a simple center-surround RF structure and may not fully capture the complexity of more diverse RF types or subcellular release specialization present in some BCs. In particular, our study did not specifically investigate the measurement of directional release from individual axonal terminals, as previously demonstrated in type 2 and 7 BCs that innervate OFF- and ON-SACs (*Matsumoto et al., 2021*). We attribute this result to the sparsity of the directionally tuned release sites, which are overshadowed by the release from non-directionally tuned units. In addition, it is presently unknown whether the contacts made by DS boutons on SACs are aligned with the soma-dendritic tip axis to enhance DS in the outward direction (*Srivastava et al., 2022*). These

aspects represent avenues for future research and could provide further insights into the mechanisms underlying DS in retinal circuits.

We were intrigued by the discovery of presynaptic cells exhibiting broad RFs that spanned hundreds of microns. Our findings bear a resemblance to a recent study that described elongated RFs in type 5A BCs (*Hanson et al., 2023*), challenging the classical understanding of bipolar RF structure, which suggests that the extent of the center component is limited to the size of their dendritic fields, typically below 50 μm in the mouse (*Euler et al., 2014*; *Franke et al., 2017*; *Gaynes et al., 2022*; *Kuo et al., 2016*; *Schwartz et al., 2012*; *Strauss et al., 2022*; *Turner et al., 2018*; *Wienbar and Schwartz, 2018*), but see *Asari and Meister, 2012* for an example of large bipolar RFs in the salamander. One plausible mechanistic explanation for the observed extensive RFs is the presence of gap-junction coupling among neighboring BCs (*Kuo et al., 2016*), which could be part of the adaptive changes in the bipolar–SAC circuits to repeated stimulation (*Ankri et al., 2020*; *Vlasits et al., 2014*). If correct, the mechanism could explain why the number of functional clusters observed in our dataset exceeds the number of BC subtypes known to connect to SACs in the mouse (five OFF and four ON types, *Ding et al., 2016*; *Greene et al., 2016*; *Kim et al., 2014*). In line with this interpretation, while most functional clusters were observed in multiple recording sessions, some clusters were observed more rarely (*Figure 6—figure supplement 1*) – perhaps due to differently adapted state of the retina in these preparations.

Another potential explanation for the abundant functional classes observed in our recordings is the possibility of glutamate release originating from amacrine cells (*Grimes et al., 2011*; *Kim et al., 2015*; *Lee et al., 2014*). It is worth noting that, to the best of our knowledge, glutamatergic amacrine cells typically do not establish direct synapses with SACs. However, these cells release glutamate in close proximity to ON-SAC dendrites (*Mani et al., 2023*). This proximity raises the possibility that the glutamatergic output from these amacrine cells could influence the iGluSnFR fluorescence observed in SACs. Glutamatergic amacrine cells are known to exhibit a small center region and pronounced surrounds (*Chen et al., 2017*). Therefore, if glutamate release from amacrine cells contributes to the signals detected in our recordings, it is plausible that it corresponds to one of the functional clusters characterized by narrow RF centers.

Mouse SACs exhibit spatial preference to different presynaptic bipolar types (*Ding et al., 2016*; *Greene et al., 2016*; *Kim et al., 2014*). However, this spatial separation is not absolute, especially in ON-SACs, where most dendritic regions may receive input from multiple presynaptic cell types. The space-time wiring model's effectiveness in conveying directional performance diminishes when a clear boundary does not exist between proximally and distally innervating presynaptic populations (*Figure 10*). It is important to note that we intentionally did not impose a strict match between the synaptic distribution in our model and biological data. As a result, our findings establish an upper limit on the contribution of the space-time wiring mechanism to SAC directional tuning. While we observed a slightly higher DS in the OFF population, consistent with a previous study (*Fransen and Borghuis, 2017*), our investigation reveals that the waveforms produced by presynaptic cells in response to moving objects do not fully exploit the potential for directional tuning that can be achieved with synthetic BCs, as depicted in *Figures 2 and 10*. It is worth highlighting that the DS levels we project for experimentally recorded glutamate signals exhibit 30-50% enhancement compared to the directional performance of a space-time wiring model observed in a recent study by *Srivastava et al., 2022* The primary distinction between our studies likely arises from the fact that *Srivastava et al., 2022* investigated flash-evoked glutamate waveforms. In contrast, we integrated over responses to moving stimuli. This discrepancy suggests that, while suboptimal, motion responses within the bipolar population still offer a more suitable foundation for achieving directional tuning. Notably, *Wu et al., 2023* put forward the idea that the spatial arrangement of midget and DB4/5 BCs in the primate retina might contribute to DS computation through a similar space-time wiring mechanism. While we lack information regarding the exact shapes of motion responses in primate BCs, it is reasonable to speculate that if they exhibit faster dynamics than those observed in mice, the space-time wiring model's influence on SAC function in primates may be even more pronounced.

These results strongly indicate that additional mechanisms are necessary to account for the observed DS capabilities in a manner consistent with empirical evidence. SACs possess intricate morphologies and express multiple voltage-gated channels (*Morrie and Feller, 2018*; *Poleg-Polsky et al., 2018*; *Yan et al., 2020*). These cells are integral components of complex neural circuits, the

architectural details of which remain incompletely understood but allow SACs to convey function-ally distinct signals to postsynaptic targets (*Pottackal et al., 2021*; *Sethuramanujam et al., 2021*). Achieving DS undoubtedly relies on a multifaceted interplay of various factors. These include, but are not limited to, the potential feedback mechanisms originating from SACs to presynaptic BCs and neighboring amacrine cells, as well as spatiotemporal modulation of postsynaptic signals by voltage-gated channels (*Ezra-Tsur et al., 2021*).

Overall, by combining experimental methods and computational modeling, our study contributes to a deeper comprehension of the bipolar organization supporting the space-time wiring model. We address the limitations of solely focusing on static stimuli and shed light on the dynamic nature of signal processing in BCs. This knowledge is crucial for elucidating the intricate mechanisms underlying visual computations in the DS circuit in the retina.

## Methods
### NEURON simulation
Multicompartmental simulations were performed using NEURON 8.2 (https://www.neuron.yale.edu/neuron) on four reconstructed SAC morphologies (https://neuromorpho.org/neuron_info.jsp?neuron_name=185exported, https://neuromorpho.org/neuron_info.jsp?neuron_name=cell8_wt_traces, https://neuromorpho.org/neuron_info.jsp?neuron_name=cell2_wt_traces, https://neuromorpho.org/neuron_info.jsp?neuron_name=cell1_wt_traces). The number of segments varied from 495 to 879. The diameter of branches further than 30 μm from the soma was set to 200 nm. The initial global passive parameters were as follows: passive conductance = $4e^{-4}$ S/cm$^2$, membrane capacitance = 1 μF/cm$^2$, reversal potential = –60 mV, and axial resistance = 150 Ω cm. N-type calcium conductance model, adapted without modifications from *Benison et al., 2001*, was distributed throughout the entire dendritic tree. In order to calculate the internal calcium levels, a calcium diffusion mechanism was incorporated into all dendrites. This mechanism had a time constant of 50 ms, and the resting calcium levels were set to 100 nM.

The SAC received innervation from 200 BC inputs, which were randomly distributed across prox-imal dendrites (within a distance of <110 μm from the soma). Each synapse was assumed to be asso-ciated with a single BC, and the RF center of the BC was aligned with the postsynaptic position. The BC RFs consisted of Gaussian-shaped components for the center and surround regions.

Stimuli were presented within a 1-mm-wide arena, matching the experimentally presented stimuli (see below). To determine RF activation, the spatial overlap between the Gaussian functions describing the center and surround RF components and the shape of the stimulus was computed separately for each time step ($\Delta t$ = 1 ms).

$Area_t$ corresponds to the normalized fraction of the stimulated RF area at time $t$, computed for each component. For full-field static flashes, the entire center and surround were activated when the stimulus was presented. For moving stimuli, $area_t$ was the sum of the area of the Gaussian function describing the center or surround RF component that spatially overlapped with the stimulus:

$$area_t = \sum_{time=1}^{t} e^{\left[-\left(time - \frac{\frac{arena\,size}{speed \times \Delta t}}{\frac{width}{2\sqrt{ln2} \times speed \times \Delta t}}\right)\right]}$$

where *width* is the FWHM of the corresponding RF component. Subsequently, RF center and surround responses at time $t$ were determined from the following equations:

$$RF_{center,t} = \left(area_{center,t} - RF_{center,t-1}\right)/risetime_{center} + RF_{center,t-1} \times adaptation_{t-1}$$

$$adaptation_t = max\left(0, adaptation_{t-1} - RF_{center,t}/decaytime_{center}\right)$$

$$RF_{surround,t} = \left(area_{surround,t} - RF_{surround,t-1}\right)/risetime_{surround} \times strength_{surround}$$

The integration of center and surround RFs was calculated in a conductance-based model of synaptic integration, determined from the excitatory and inhibitory driving forces:

$$RF_{full,t} = RF_{amplitude} \times \frac{RF_{center,t} + RF_{surround,t} \times reversal_{surround}}{RF_{center,t} + RF_{surround,t} + R}$$

where $R_{in}$ is the input resistance. Because the magnitude of RF activation depended on its size, $R_{in}$ was set to be proportional to the spatial extent of the center.

$$R_{in} = 0.1 \times area_{center,\infty}$$

RF activation in each BC was temporally adjusted based on stimulus velocity and the spatial position:

$$\Delta t_{shift} = \frac{RF_x}{Velocity}$$

The following constraints were imposed: RF amplitude, surround strength, and reversal were permitted to vary between 0 and 1; rise/decay times and RF widths were positive. Overall, $RF_{full}$ was expressed in unitless units in the [0,1] interval, with values closer to 1 signifying strong activation.

## Training of evolutionary models

During the training of evolutionary models aimed at understanding the contribution of BC dynamics to SAC DS (as shown in *Figures 2–4*), the bipolar population was equally divided into proximally and distally innervating cells according to their proximity to SAC soma.

For each generation, a total of 16 bipolar–SAC models were executed simultaneously on a Dell Precision 5820 workstation, with one model per processor thread. In the first generation, random values were assigned to seed each network model. A separate RF description was instantiated with random values within the specified limits for the two BC populations.

Likewise, random initial values were chosen for passive conductance (ranging from $1e^{-5}$ to $1e^{-3}$ S/cm$^2$), axial resistance (constrained between 50 and 300 $\Omega$ cm), N-type calcium conductance and voltage offset (limits = ± 30 mV), as well as synaptic conductance (limits = 0.01–1 nS). All models were presented with the same stimuli, consisting of bars moving from left to right and from right to left at five different speeds: 0.25, 0.5, 1, 2, and 4 mm/s. The height of the bar was 1 mm, its duration was 2 s. The intensity of the stimuli was set to 1 and the background was set to zero (AU).

Calcium signals were recorded from SAC sites located on the distal dendrites, specifically those within proximity of less than 30 μm from the horizontal axis. When a single branch was stimulated, only sites on that dendrites were included in the analysis. For each stimulation speed (comprising two different stimulation directions), we computed the DS index of each site as follows:

$$DSI = \frac{R_{Outward} - R_{Inward}}{R_{Outward} + R_{Inward}}$$

where $R_{Outward}$ and $R_{Inward}$ are the peak calcium levels recorded for the outward and inward directions from the perspective of the dendrite. To minimize extreme calcium signals, we computed the directional metric:

$$directional\,metric = DSI \times e^{\left[\frac{-(R_{Outward} - Ca_{opt})^2}{(Ca_{opt})^2}\right]}$$

where $Ca_{opt}$ was the optimal calcium level in the outward direction, set to 500 nM.

Subsequently, the models were ranked based on the average directional metric calculated over the five stimulation speeds. The two best-performing models were retained without any changes, while the parameters of the rest of the models were randomly selected to match either the best or second-best-performing models and then mutated as follows: for each parameter describing the presynaptic RF and postsynaptic SAC properties, a Gaussian distribution with a mean of 1 and a standard deviation of 5% was used to determine a scaling factor. This scaling factor was multiplied with the parameter value and combined with a random value drawn from a uniform distribution ranging from –0.015 to 0.015. These modified (mutated) models constituted the next generation of candidate solutions. Typically, we evolved the model over 100 generations, as empirical evidence has shown that increasing the number of generations beyond this point does not yield significantly improved results.

## RF estimation from recorded glutamate waveforms

To estimate the RF that could mediate the recorded glutamate waveforms presented in *Figure 7* and *Figure 7—figure supplement 1*, we employed an EA for training RF models to match the shape of experimentally observed glutamate release clusters. Each cluster was considered separately during the analysis.

The model contained a single RF, and its parameters were described using the same parameter set as mentioned earlier. It is important to note that the fluorescent signal produced by iGluSnFR represents a temporally filtered version of the original glutamatergic drive (*Armbruster et al., 2020*; *Hain and Moser, 2023*; *Srivastava et al., 2022*). In order to mimic the iGluSnFR waveforms in the models, we applied a filtering process to the output of the simulation. On each time step $t_0$, we convolved the value of the simulated response with a wavelet described as the difference between two exponential functions:

$$filter_{t0} = \left( e\left| \frac{t - t0}{F_{decay}} - e^{\frac{t - t0}{F_{rise}}} \right. \right)$$

where $F_{rise}$ = 10 ms, $F_{decay}$ = 50 ms are the rise and decay times of the filter, respectively. The resulting vectors starting at time $t_0$ and lasting till the end of simulation duration were combined to produce the filtered version of the full response.

Following the initial random instantiation, simulated RF responses were calculated for the five different speeds. The fitness of each model was then evaluated based on the mean square error between the simulated waveforms and the experimentally recorded waveforms. The next generation was formed by introducing potentially mutated offspring of models that exhibited the lowest errors. These mutations were performed according to the methodology described above.

## Determination of optimal DS with experimentally recorded glutamatergic waveforms

To investigate the impact of the spatial dependence of the presynaptic innervation waveform on post-synaptic DS, we conducted an initial analysis using a modified version of the bipolar–SAC model. In this derivative model, we systematically replaced the RF descriptions of proximal and distal BCs with fits to experimental data. The focus of the evolution process was on altering postsynaptic parameters to optimize synaptic integration toward the largest directional performance. For the data shown in *Figure 9*, we employed a different training approach. We divided the input synapses into annuli with a width of 10 µm, centered around the soma. Initially, each annulus was randomly assigned one of the RF fits corresponding to functional clusters.

As the model evolved, its goal was to achieve the best possible directional performance based on the postsynaptic parameters. Additionally, there was a 5% probability for each annulus to replace the assigned functional cluster identity. This allowed for evolution in the spatial configuration of the functional clusters, enabling us to examine what spatial distribution of experimentally recorded waveforms produces the optimal DS.

## Virus expression and imaging procedures

All animal procedures were conducted in accordance with US National Institutes of Health guidelines, as approved by the University of Colorado Institutional Animal Care and Use Committee (IACUC). Mice were housed in a 12 light/12 dark cycle at room temperature (~22°C), 40–60% relative humidity. For intravitreal virus injections, 8–12-week-old ChAT-Cre transgenic mice (Jax strain 031661, https://www.jax.org/strain/031661) were anesthetized with isoflurane; ophthalmic proparacaine and phenylephrine were applied for pupil dilation and analgesia. A small incision at the border between the sclera and the cornea was made with a 30-gauge needle. Then, 1 µl of AAV9.hsyn.FLEX.iGluSnFR. WPRE.SV40 (a gift from Loren Looger, Addgene plasmid # 98931; http://n2t.net/addgene:98931; RRID:Addgene_98931, $10^{13}$ vg/ml in water) solution was injected with a blunt tip (30 gauge) modified Hamilton syringe (https://www.borghuisinstruments.com/). Experiments on retinas from all animal groups were performed 2–6 wk following virus injection on 11–17-week-old animals (four males and four females).

Mice were not dark-adapted to reduce rod-pathway activation. Then, 2 hr after enucleation, retina sections were whole mounted on a platinum harp with their photoreceptors facing down, suspended ~1 mm above the glass bottom of the recording chamber. The retina was kept at ~32°C and continuously perfused with Ames media (Sigma-Aldrich, https://www.sigmaaldrich.com/) equilibrated with 95% $O_2$/5% $CO_2$.

## Visual stimulation

Light stimuli were generated in Igor Pro 8 (Wavemetrics, https://wavemetrics.com/) running in Windows 10 and displayed with a DPL projector (Texas Instruments, https://www.ti.com/, model 4710EVM-G2) connected as a second monitor. Only the blue projector LED was used, and its light was further filtered with a 450 nm low-pass filter (Thorlabs, https://thorlabs.com/, FEL0450). Light from the visual stimulus was focused by the condenser to illuminate the tissue at the focal plane of the photoreceptors (resolution = 10 μm/pixel, background light intensity = 30,000–60,000 R* rod$^{-1}$). Both vertical and horizontal light stimulus positions were checked and centered daily before the start of the experiments. The following light stimulus patterns were used: static flash covering the entire display (1000 × 1000 μm) presented for 2–4 s. A 1-mm-long bar moving either to the left or the right directions (speeds = 0.25, 0.5, 1, 2, 4 mm/s; dwell time over each pixel = 2 s). These stimuli were repeated three times. Typically, stimulus contrast was set to 60% Michelson contrast. To record glutamate signals to OFF-SACs, we reversed the intensity of the moving bar stimulus and the background. In some experiments, we masked different portions of the display to remain at background light levels (*Gaynes et al., 2022*). To map the spatial RFs, we flashed 10 × 1000 μm bars for 200 ms every 400 ms. The bars were presented over five evenly spaced (36°) orientations in a pseudorandom sequence over 32 spatial positions in every orientation to densely cover ~300 μm of visual space centered on the imaged region.

## Recording procedures

Glutamate imaging was performed with Throlabs Bergamo galvo-galvo two-photon microscope using the Thorimage 4.1 acquisition software (Throlabs). A pulsed laser light (920 nm, ~1 μW output at the objective; Chameleon Ultra II, Coherent, https://www.coherent.com/) was used for two-photon excitation projected from an Olympus 20× (1 NA) objective. A descanned (confocal) photomultiplier tube (PMT) was used to acquire fluorescence between 500 and 550 nm. The confocal pinhole (diameter = 1 mm) largely prevented stimulus light (focused on a different focal plane), from reaching the PMT, allowing us to present the visual stimulus during two-photon imaging. A photodiode mounted under the condenser sampled transmitted laser light to generate a reference image of the tissue. Fluorescence signals were collected in a rapid bidirectional frame scan mode (128 × 64 pixels; ~50 Hz). The line spacing on the vertical axis was doubled to produce a rectangular imaging window (~164 × 164 μm in size; the corresponding pixel size was 1.28 μm). To reduce shot noise, images were subsampled by averaging 2 × 2 neighboring pixels and filtered by a 20 Hz low-pass filter offline. Horizontal and vertical image drifts were corrected online using a reference z-stack acquired before time-series recordings.

Labeled cells in the Chat-Cre/tdTomato line (Jax strain 007909, https://www.jax.org/strain/007909) were targeted for whole-cell recordings using 4–8 MΩ pipettes filled with 90 mM $CsCH_3SO_4$, 20 mM TEA-Cl, 10 mM HEPES, 10 mM EGTA, 10 mM phosphocreatine disodium salt hydrate, 5 mM QX-314, 4 mM Mg-ATP, and 0.4 mM Na-GTP. Recordings were obtained in voltage-clamp configuration with a Double IPA amplifier (Sutter, https://www.sutter.com/), low-pass filtered at 2 kHz using a custom acquisition software written in Igor Pro.

## Analysis

The fluorescence signals were averaged across multiple presentations of the visual protocol using Igor Pro 8. Specifically, pixels with dF/F values >20% were selected for subsequent clustering analysis, which involved two steps.

In the first step, the extent of contiguous ROIs with similar response kinetics was determined. This was achieved on a pixel-by-pixel basis by first averaging the fluorescence signals across repeated presentations of the stimuli. Then, the pixel's response shape was measured within a 1 s window centered around the time when the bar motion reached the center of the display during the 10 motion trials, which included five speeds and two directions. We applied hierarchical clustering to cluster the

responses into ROIs based on the similarity of pixels' waveforms in these windows using the built-in Igor Pro 'FPClustering' function (*Gaynes et al., 2022*). The number of resulting ROIs was adjusted until no ROI spanned more than 10 µm in size. The ROIs were manually curated, and any ROIs with pixel variability exceeding a coefficient of variation threshold of 1 were removed from further analysis.

In the second step, we combined individual ROIs from multiple scan fields and mice into functional clusters. This process was separate for ON- and OFF-SACs. For each ROI, we computed the mean motion response shape by aligning the motion responses to both directions of stimulation by the half-maximum rise time. The alignment was performed as follows: first, we calculated the half-maximum rise times for glutamate signals recorded in response to left and right-moving stimuli. Next, we determined the mean of these rise times, which served as a reference point for alignment. In the final step, the response waveforms to both directions of stimulation were shifted to the reference point. The magnitude of the shift was exactly half of the mean time calculated in the previous step. Thus, the alignment procedure produced a single trace, whose half-maximum rise time was precisely in between the rise times for the responses to the two directions. The rise time of the aligned signal does not depend on the position of the RF (*Gaynes et al., 2022*), allowing us to compare the waveforms of glutamate response across different locations and recordings. Secondary hierarchical clustering was performed in R v3.6 (R Foundation for Statistical Computing). Motion responses to 0.5 mm/s stimuli were first converted to data frames in R. The traces were smoothed with a combination of a Butterworth (filter order = 1, critical frequency = 0.1) and rolling average filters, baseline subtracted, and normalized to the peak amplitude. For the ON responses, we discarded time points with variance less than 0.5 as this aided with convergence to biologically plausible clusters. To quantify the number of functional clusters, we used within-cluster variance (*Warren Liao, 2005*). Within-cluster variance is computed by finding the sum of the squared differences of each observation in a proposed cluster and the mean of that cluster. We used automated assessment of the elbow plots (within-cluster variance as a function of the number of clusters) to identify optimal cluster numbers for a grid search of distance computations and clustering algorithms available in base R and the dynamic time warping library. We then used expert assessment to identify the most plausible clustering from this small set of options (*Figure 6—figure supplement 1*). Empirically, we found that the ward.D2 hierarchical clustering method, available in the base R 'hclust' function, provides the clearest cluster separation. We used the Euclidean and maximum distances for the ON- and OFF-SAC datasets, respectively.

Tests of statistical significance were performed in Igor Pro using built-in functions.

RF mapping was determined from a two-dimensional Gaussian fit to the responses to the oriented bars, as described previously (*Johnston et al., 2014*; *Poleg-Polsky et al., 2018*). The FWHM of the RF was calculated along the axis of motion from the spatial time constant generated by the fit ($\sigma_x$) as follows:

$$FWHM = 2\sqrt{2ln2}\sigma_x$$

The transiency index (TI) was calculated as the ratio between the peak and the mean of the response within the stimulation window. TI = 1 indicates a sharp and transient response, TI close to zero is produced by sustained plateaus.

IPL depth was measured from the transmitted light channel extracted from the z-stack taken of the entire width of the retina that accompanied all functional recordings. The curvature of the retina was corrected by measuring the height of the inner limiting membrane at the four corners and the center of the image stack and fitting a curved plane that crossed these five points.

## Acknowledgements

This work was supported by NIH grant (R01 EY030841) to AP-P.

# Additional information

## Funding

| Funder | Grant reference number | Author |
|---|---|---|
| National Institutes of Health | R01 EY030841 | John A Gaynes |

The funders had no role in study design, data collection and interpretation, or the decision to submit the work for publication.

## Author contributions

John A Gaynes, Michael J Grybko, Investigation, Writing - review and editing; Samuel A Budoff, Software, Formal analysis, Writing - review and editing; Alon Poleg-Polsky, Conceptualization, Data curation, Software, Formal analysis, Supervision, Funding acquisition, Visualization, Methodology, Writing - original draft, Project administration

## Author ORCIDs

Alon Poleg-Polsky ⓘ http://orcid.org/0000-0003-1327-5129

## Ethics

All animal procedures were conducted in accordance with US National Institutes of Health guidelines, as approved by the University of Colorado Institutional Animal Care and Use Committee (IACUC) protocol #0348.

Reviewer #1 (Public Review): https://doi.org/10.7554/eLife.90456.3.sa1
Reviewer #2 (Public Review): https://doi.org/10.7554/eLife.90456.3.sa2
Reviewer #3 (Public Review): https://doi.org/10.7554/eLife.90456.3.sa3
Author Response https://doi.org/10.7554/eLife.90456.3.sa4

# Additional files

## Supplementary files
• MDAR checklist

## Data availability

The code for the visual stimulation and simulations is available at https://github.com/PolegPolskyLab/DS_Bipolar_Inputs_SAC/ (copy archived at *Poleg-Polsky, 2023*).

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
