## [Editor Report · eLife assessment]

This **important** study uses a combination of computational modeling and glutamate imaging to show how a particular synaptic organization referred to as space-time wiring contributes minimally to a dendritic computation that occurs in the retina. The evidence supporting the claims of the authors is **compelling**, incorporating new findings regarding dynamic receptive field properties, an improvement over previous modeling and experimental results based on static visual stimuli. The work will be of interest to retinal neurobiologists and neurophysiologists interested in dendritic computations.

---

## [Referee Report · Reviewer #1 (Public Review)]

Summary:

Direction selectivity (DS) in the visual system is first observed in the radiating dendrites of starburst amacrine cells (SACs). Studies over the last two decades have aimed to understand the mechanisms that underlie these unique properties. Most recently, a 'space-time' model has garnered special attention. This model is based on two fundamental features of the circuit. First, distinct anatomical types of bipolar cells (BCs) are connected to proximal/distal regions of each of the SAC dendritic sectors (Kim et al., 2014). Second, that input across the length of the starburst is kinetically diverse, a hypothesis that has only recently gained some experimental support using iGluSnFR imaging (Srivastava et al., 2022). However, in these prior studies, the sustained/transient distinctions in BC input that are proposed to underlie direction selectivity were shown to be present mainly in responses to stationary stimuli. When BC receptive field properties are probed using white noise stimuli, the kinetic differences between proximal/distal BC input are relatively subtle or nonexistent (Gaynes et al., 2022; Strauss et al., 2022, Srivastava et al., 2022). Thus, if and how BCs contribute to direction selectivity driven by moving spots that are commonly used to probe the circuit remains to be clarified. To address this issue, Gaynes et al., combine evolutionary computational modeling (Ankri et al., 2020) with two-photon iGluSnFR imaging to address to what degree BCs contribute to the generation of direction selectivity in the starburst dendrites.

Strengths:

Combining theoretical models and iGluSnFR imaging is a powerful approach as it first provides a basic intuition on what is required for the generation of robust DS, and then tests the extent to which the experimentally measured BC output meets these requirements.

The conclusion of this study builds on the previous literature and comprehensively considers the diverse BC receptive field properties that may contribute to DS (e.g. size, lag, rise time, decay time).

By 'evolving' bipolar inputs to produce robust DS in a model network, these authors provide a sound framework for understanding which kinetic properties could potentially be important for driving downstream DS. They suggest that response delay/decay kinetics, rather than the center/surround dynamics are likely to be most relevant (albeit the latter could generate asymmetric responses to radiating/looming stimuli).

Weaknesses:

Finally, these authors report that the experimentally measured BC responses are far from optimal for generating DS. Thus, the BC-based DS mechanism does not appear to explain the robust DS observed experimentally (even with mutual inhibition blocked). Nevertheless, I feel the comprehensive description of BC kinetics and the solid assessment of the extent to which they may shape DS in SAC dendrites, is a significant advancement in the field.

---

## [Referee Report · Reviewer #2 (Public Review)]

Summary:

In this study, the authors sought to understand how the receptive fields of bipolar cells contribute to direction selectivity in starburst amacrine cell (SAC) dendrites, their post synaptic partners. In previous literature, this contribution is primarily conceptualized as the 'space-time wiring model', whereby bipolar cells with slow-release kinetics synapse onto proximal dendrites while bipolar cells with faster kinetics synapse more distally, leading to maximal summation of the slow proximal and fast distal depolarizations in response to motion away from the soma. The space-time wiring contribution to SAC direction selectivity has been extensively tested in previous literature using connectomic, functional, and modeling approaches. However, the authors argue that previous functional studies of bipolar cell kinetics have focused on static stimuli, which may not accurately represent the spatiotemporal properties of the bipolar cell receptive field in response to movement. Moreover, this group and others have recently shown that bipolar cell signal processing can change directionally when visual stimuli starts within the receptive field rather than passing through it, complicating the interpretation of moving stimuli that start within a bipolar cell of interest's receptive field (e.g. stimulating only one branch of a SAC or expanding/contracting rings). Thus, the authors choose to focus on modeling and functionally mapping bipolar cell kinetics in response to moving stimuli across the entire SAC dendritic field.

General Comments:

There have been several studies that have addressed the contribution of space-time wiring to SAC process direction selectivity. This study offers a more complete assessment of potential impact space-time wiring can have on this dendrite computation. The experimental results based on glutamate imaging assess the kinetics of glutamate release under conditions of visual stimulation across a large region of retina largely confirm previous observations. By combining their model with this experiment data, they conclude that even the optimal space-time wiring is not sufficient to explain the SAC process DS. Though there is no conclusion which of the many other proposed cellular and circuit mechanisms could potentially contribute to this computation, the limited role for spacetime wiring is firmly established.

---

## [Referee Report · Reviewer #3 (Public Review)]

Summary:

Gaynes et al. investigated the presynaptic and postsynaptic mechanisms of starburst amacrine cell (SAC) direction selectivity in the mouse retina by computational modeling and glutamate sensitivity (iGluSnFR) imaging methods. Using the SAC computational simulation, the authors initially tested bipolar cell contributions (space-time wiring model, presynaptic effect) and SAC axial resistance contributions (postsynaptic effect) to the SAC DS. Then, the authors conducted two-photon iGluSnFR imaging from SACs to examine the presynaptic glutamate release and found seven clusters of ON-responding and six clusters of OFF-responding bipolar cells. They were categorized based on their response kinetics: delay, onset phase, decay time, and others. Finally, the authors used cluster data to reconstruct bipolar cell inputs to SACs that generate direction selectivity. They concluded that presynaptic effects through the space-time wiring model only account for a fraction of SAC DS.

The article has many interesting findings, and the data presentation is superb. Strengths and weaknesses are summarized below.

Major Strengths:

The authors utilized solid technology to conduct computational modeling with Neuron software and a machine-learning approach based on evolutionary algorithms. Results are effectively and thoroughly presented.

The space-time wiring model was evaluated by changing bipolar cell response properties in the proximal and distal SAC dendrites. Many response parameters in bipolar cells are compared, and DSI is compared in Figure 3. These parameter comparisons are valuable to the field.

Two-photon microscopy was used to measure the bipolar cell glutamate outputs onto SACs by conducting iGluSnFR imaging. All the data sets, including images and transients, are elegantly presented. The authors analyzed the response based on various parameters, which generated more than several response clusters. The clustering is convincing.

Major Weaknesses:

The computational modeling demonstrates intriguing results: SAC dendritic morphology produces dendritic isolation, and a massive input overcomes the dendritic isolation (Figure 1). This modeling seems to be generated by basic dendritic cable properties. However, it has been reported that SAC dendrites express Kv3 and voltage-gated Ca channels. Are they incorporated into this model? If not, how about comparing these channel contributions?

In Figure 9 the authors generated the bipolar cell cluster alignment based on the space-time wiring model. The space-time wiring model has been proposed based on the EM study that distinct types of bipolar cells synapse on distinct parts of SAC dendrites (Green et al 2016, Kim et al 2014). While this is one of the representative Reicardt models, it is not fully agreed upon in the field (see Stincic et al 2016). Therefore, the authors' approach might be only hypothetical without concrete evidence for geographical cluster distributions. Is there any data suggesting each cluster's location on the SAC dendrites? I assume that the iGluSnFR imaging was conducted on the SAC dendritic network, which does not provide geographical information. How about injecting the iGluSnFR-AAV at a lower titer, which labels only some SACs in a tissue? This method may reveal each cluster's location on SAC dendrites.

The authors found that there are seven ON clusters and six OFF clusters, which are supposed to be bipolar cell terminals. However, bipolar cells reported to provide synaptic inputs are T-7, T-6, and multiple T-5s for ON SACs and T-1, T-2, and T-3s for OFF SACs. The number of types is less than the number of clusters. Is there a possibility of clusters belonging to glutamatergic amacrine cells? Please provide a discussion regarding the relations between clusters and cell types.

In Figure 5B, representative traces are shown responding to moving bars in horizontal directions. These did not show different responses to two directional stimuli. Is there any directional preference from other ROIs? Yonehara's group recently exhibited the bipolar cells' direction selectivity (Matsumoto et al 2021). Did you see any correlations with their results? Please discuss.

---

## [Author Response]

The following is the authors’ response to the current reviews.

For the final Version of Record the following changes will be included:

1. Figure 4: Example traces replaced with a more representative simulation run that is more similar to the mean.

2. Methods: Description of the alignment procedure expanded to explain the algorithm steps better.

The following is the authors’ response to the previous reviews

We are grateful for the positive and insightful feedback from the editors and reviewers. These constructive comments have contributed to the enhancement of our work. We have revised the manuscript, addressing each of the comments raised. In addition, based on the commentary provided, we have introduced two new figures that offer a deeper understanding of our research findings:

In new Figure 7, we present the analysis of the difference in onset times between motion and flash responses. This figure also includes a simple illustration elucidating the origins of these differences, highlighting the varying engagement of receptive fields by these stimuli. The data presented in this figure were initially featured in the main text of the original manuscript.Figure 11 offers a detailed comparison of the temporal and spatial characteristics of the synthetic presynaptic signals driving optimal DS in SACs. We compare these characteristics with the properties extracted from recorded glutamate release. Our analysis suggests that the sluggish dynamics observed in biological signals impede effective directional integration.Below are the detailed point-by-point responses to reviewers comments.

**Reviewer #1 (Public Review):**
Summary:Direction selectivity (DS) in the visual system is first observed in the radiating dendrites of starburst amacrine cells (SACs). Studies over the last two decades have aimed to understand the mechanisms that underlie these unique properties. Most recently, a 'space-time' model has garnered special attention. This model is based on two fundamental features of the circuit. First, distinct anatomical types of bipolar cells (BCs) are connected to proximal/distal regions of each of the SAC dendritic sectors (Kim et al., 2014). Second, that input across the length of the starburst is kinetically diverse, a hypothesis that has been only recently demonstrated experimentally using iGluSnFR imaging (Srivastava et al., 2022). However, the stark kinetic distinctions, i.e., the sustained/transient nature of BC input to SACs dendrites appear to be present mainly in responses to stationary stimuli. When BC receptive field properties are probed using white noise stimuli, the kinetic differences between BCs are relatively subtle or nonexistent (Gaynes et al., 2022; Strauss et al., 2022, Srivastava et al., 2022). Thus, if and how BCs contribute to direction selectivity driven by moving spots that are commonly used to probe the circuit remains to be clarified. To address this issue, Gaynes et al., combine evolutionary computational modeling (Ankri et al., 2020) with two-photon iGluSnFR imaging to address to what degree BCs contribute to the generation of direction selectivity in the starburst dendrites in response to stimuli that are commonly used experimentally.Strengths:Combining theoretical models and iGluSnFR imaging is a powerful approach as it first provides a basic intuition on what is required for the generation of robust DS, and then tests the extent to which the experimentally measured BC output meets these requirements.The conclusion of this study builds on the previous literature and comprehensively considers the diverse BC receptive field properties that may contribute to DS (e.g. size, lag, rise time, decay time).By 'evolving' bipolar inputs to produce robust DS in a model network, these authors provide a sound framework for understanding which kinetic properties could potentially be important for driving downstream DS. They suggest that response delay/decay kinetics, rather than the center/surround dynamics are likely to be most relevant (albeit the latter could generate asymmetric responses to radiating/looming stimuli).Weaknesses:Finally, these authors report that the experimentally measured BC responses are far from optimal for generating DS. Thus, the BC-based DS mechanism does not appear to explain the robust DS observed experimentally (even with mutual inhibition blocked). Nevertheless, I feel the comprehensive description of BC kinetics and the solid assessment of the extent to which they may shape DS in SAC dendrites, is a significant advancement in the field.
**Reviewer #2 (Public Review):**
Summary:In this study, the authors sought to understand how the receptive fields of bipolar cells contribute to direction selectivity in starburst amacrine cell (SAC) dendrites, their post synaptic partners. In previous literature, this contribution is primarily conceptualized as the 'space-time wiring model', whereby bipolar cells with slow-release kinetics synapse onto proximal dendrites while bipolar cells with faster kinetics synapse more distally, leading to maximal summation of the slow proximal and fast distal depolarizations in response to motion away from the soma. The space-time wiring contribution to SAC direction selectivity has been extensively tested in previous literature using connectomic, functional, and modeling approaches. However, the authors argue that previous functional studies of bipolar cell kinetics have focused on static stimuli, which may not accurately represent the spatiotemporal properties of the bipolar cell receptive field in response to movement. Moreover, this group and others have recently shown that bipolar cell signal processing can change directionally when visual stimuli starts within the receptive field rather than passing through it, complicating the interpretation of moving stimuli that start within a bipolar cell of interest's receptive field (e.g. stimulating only one branch of a SAC or expanding/contracting rings). Thus, the authors choose to focus on modeling and functionally mapping bipolar cell kinetics in response to moving stimuli across the entire SAC dendritic field.General CommentsThere have been several studies that have addressed the contribution of space-time wiring to SAC process direction selectivity. The impact of this project is to show that this contribution is limited. First, the optimal solution obtained by the evolutionary algorithm to generate DS processes is slow proximal and fast distal inputs - exactly what is predicted by space-time wiring, which is exactly what is required of the HRC model. Hence, this result seems expected and it's not clear what the alternative hypothesis is. Second, the experimental results based on glutamate imaging to assess the kinetics of glutamate release under conditions of visual stimulation across a large region of retina confirm previous observations but were important to test. Third, by combining their model model with this experiment data, they conclude that even the optimal space-time wiring is not sufficient to explain the SAC process DS. The results of this approach might be more impactful if the authors come to some conclusion as to what factors do determine the direction selectivity of the SAC process since they have argued that all the current models are not sufficient.
**Reviewer #3 (Public Review):**
Gaynes et al. investigated the presynaptic and postsynaptic mechanisms of starburst amacrine cell (SAC) direction selectivity in the mouse retina by computational modeling and glutamate sensitivity (iGluSnFR) imaging methods. Using the SAC computational simulation, the authors initially tested bipolar cell contributions (space-time wiring model, presynaptic effect) and SAC axial resistance contributions (postsynaptic effect) to the SAC DS. Then, the authors conducted two-photon iGluSnFR imaging from SACs to examine the presynaptic glutamate release, and found seven clusters of ON-responding and six clusters of OFF-responding bipolar cells. They were categorized based on their response kinetics: delay, onset phase, decay time, and others. Finally, the authors generated a model consisting of multiple clusters of bipolar cells on proximal and distal SAC dendrites. When the SAC DS was measured using this model, they found that the space-time wiring model accounted for only a fraction of SAC DS.The article has many interesting findings, and the data presentation is superb. Strengths and weaknesses are summarized below.Major Strengths:• The authors utilized solid technology to conduct computational modeling with Neuron software and a machine-learning approach based on evolutionary algorithms. Results are effectively and thoroughly presented.• The space-time wiring model was evaluated by changing bipolar cell response properties in the proximal and distal SAC dendrites. Many response parameters in bipolar cells are compared, and DSI was compared in Figure 3.• Two-photon microscopy was used to measure the bipolar cell glutamate outputs onto SACs by conducting iGluSnFR imaging. All the data sets, including images and transients, are elegantly presented. The authors analyzed the response based on various parameters, which generated more than several response clusters. The clustering is convincing.Major Weaknesses:• In Figure 9, the authors generated the bipolar cell cluster alignment based on the space-time wiring model. The space-time wiring model has been proposed based on the EM study that distinct types of bipolar cells synapse on distinct parts of SAC dendrites (Green et al 2016, Kim et al 2014). While this is one of the representative Reicardt models, it is not fully agreed upon in the field (see Stincic et al 2016). While the authors' approach of testing the space-time wiring model and conclusions is interesting and appreciated, the authors could address more issues: mainly two clusters were used to generate the model, but more numbers of clusters should be applied. Although the location of each cluster on the SAC dendrites is unknown, the authors should know the populations of clusters by iGluSnFR experiments. Furthermore, the authors could provide more suggestive mechanisms after declining postsynaptic factors and the space-time wiring model.

The reviewer is correct that the proximal and more distal SAC dendrites sample from different IPL depths. It should be theoretically possible to match the functional clusters we measured with anatomical bipolar cell identities. However, the stratifications of these cells have significant overlaps (Figure 6-S2), and previous attempts to match iGluSnFR signals to anatomy proved to be challenging (Franke et al., 2017; Gaynes et al., 2022; Matsumoto et al., 2019; Srivastava et al., 2022; Strauss et al., 2022). In the revised version of the manuscript, we reorder the functional clusters based on their transiency, which has a higher correlation to stratification depth (Franke et al., 2017).

We have examined a scenario in which the presynaptic population comprises more than two clusters. We constructed synthetic models whose input structure was as in Figure 10 (old Figure 9). The optimal configuration for the most proximal and distal inputs closely resembled the proximal-distal model reported in Figure 2. However, we observed a nearly linear variation in the shape of the optimal mid-range inputs, transitioning from proximal-like to distal-like responses as the distance increased. We consider this outcome to be expected based on the structure of the space-time wiring model (Kim et al., 2014). Interestingly, this was not the case with models incorporating physiologically recorded signals. As we show in Figure 10, the most common optimal directional tuning was seen when the bipolar drive consisted of two main populations, both in the ON and OFF SACs.

Finally, we believe that uncovering additional mechanisms that underlie directional selectivity in SACs represents a crucial challenge for the field to tackle. It is highly probable that achieving directional selectivity involves a complex interplay of multiple factors. This includes the organization of the presynaptic circuit, which we have partially addressed in this study, as well as the influence of postsynaptic active conductances and feedback loops involving other SACs and presynaptic cells. We have expanded the discussion section to describe the possible mechanisms

• The computational modeling demonstrates intriguing results: SAC dendritic morphology produces dendritic isolation, and a massive input overcomes the dendritic isolation (Figure 1). This modeling seems to be generated by basic dendritic cable properties. However, it has been reported that SAC dendrites express Kv3 and voltage-gated Ca channels. It seems to be that these channels are not incorporated in this model.

The reviewer's observation is accurate; the model depicted in Figure 1 did not include voltage-gated channels. Our goal was to study electrotonic isolation, which is often measured in passive models. However, while we did not incorporate voltage-gated potassium channels implicitly in the models, our simulations are rooted in previous models that were fine-tuned using empirical data. As potassium channels are expected to influence the experimentally recorded input resistance, we have indirectly accounted for their impact on the interdendritic signal propagation.

In subsequent model iterations, we have integrated voltage-gated calcium channels into our simulations to assess the signal responsible for driving synaptic release. We show that nonlinear voltage dependence of the calcium currents enhances compartmentalization of the local calcium levels (Figure 2), but did not significantly influence local voltages. Therefore, calcium channels do not appear to have a major impact on electrotonic distances.

• In Figure 5B, representative traces are shown responding to moving bars in horizontal directions. These did not show different responses to two directional stimuli. It is unclear whether directional preference was not detected, which was shown by Yonehara's group recently (Matsumoto et al 2021). Or that was not investigated as described in the Discussion.

Indeed, we observed no discernible directional differences in bipolar responses. This phenomenon can be primarily attributed to the fact that the signals originating from the limited number of directionally-tuned release sites are overshadowed by the release from non-directionally-tuned units (Matsumoto et al., 2021). In the revised discussion, we have acknowledged this limitation in our recorded data.

• The authors found seven ON clusters and six OFF clusters, which are supposed to be bipolar cell terminals. However, bipolar cells reported to provide synaptic inputs are T-7, T-6, and multiple T-5s for ON SACs and T-1, T-2, and T-3s for OFF SACs. The number of types is less than the number of clusters. Potentially, clusters might belong to glutamatergic amacrine cells. These points are not fully discussed.

We have expanded the discussion section to address these points.

**Reviewer #1 (Recommendations For The Authors):**
Major comments1. One of the main conclusions of this study is that diverse BC kinetics contribute to DS (Fig. 9). The authors nicely demonstrate using modeling that the experimentally measured BC kinetics are far from ideal. However, this conclusion is based on a model that almost exclusively relies on just two of the 7 putative BC types (e.g., C1 & C6 for On SACs) placed optimally along the dendrites, which raises two important caveats.First, given that other BC types are likely to contribute, the effects of two distinct types are likely to be diluted. Thus, the contribution of BCs to DS is likely to be significantly overestimated. Second, given that the dendrites of 10-30 SACs cross each point in the honeycomb, for the given model to work, each BC would need to connect extremely selectively to SACs. i.e., at a given point, a sustained input must only connect to the more proximal dendritic segments, while avoiding entirely the distal segments of overlapping SAC dendrites. Thus, their model requires extremely selective wiring for which there is no evidence. In fact, there is evidence to the contrary provided by Ding et al. 2016, which showed that the type 7 (proximally biased) and type 5 (distally biased) populations had a substantial overlap (assuming these BC types correspond to kinetically diverse clusters).

We wholeheartedly concur with the reviewer's perspective that our findings have led to an overestimation of the space-time wiring mechanism's role in SAC directional selectivity (DS). We have adjusted our discussion to emphasize this point. In light of this, our assertion that, even with the most favorable distribution of synaptic inputs, the space-time wiring model still does not fully account for the experimentally-determined directional tuning in SAC, remains valid.

With regard to the model, it would also be worth comparing results to previous starburst models (e.g., Tukker et al,. 2004), which demonstrated a robust DS in SAC dendrites in the absence of kinetically diverse BC input. Why is the cell-intrinsic DS so weak in the present model?

We have directly explored this question in the synthetic model (Figures 2, 3). Despite variances in the anatomy of SACs and the distribution of bipolar inputs between our model and the study by (Tukker et al., 2004), we observed remarkably similar levels of directional selectivity index computed from the voltage response (approximately 10%, as shown in Figure 3, 'Identical BCs').

The primary distinction emerged in the degree of DS amplification mediated by calcium currents. Tukker et al., 2004 reported considerably higher DS compared to our findings, despite employing similar formulations for voltage-gated calcium channel models. The key factor driving this difference lies in the fact that Tukker et al., 2004 measured amplification in proximity to the threshold of calcium channel activation. Even minor variations in membrane potentials near this threshold can lead to substantial differences in calcium influx, especially when outward stimulation results in a calcium spike. In fact, recently, Robert Smith’s group revisited the threshold-based mechanism and concluded that it often fails to produce robust DS due to the heterogeneity of membrane potentials among different terminal dendrites (Wu et al., 2023).

Our models were trained on five different stimuli velocities whose synaptic integration produced substantially different peak amplitudes. Consequently, the spike threshold alone couldn't reliably distinguish between inward and outward directions across all five conditions, resulting in reduced directional performance in our simulations. In the revised Figure 2-S2 we directly explore the performance of the model with identical BC formulations, trained on a single velocity. We find a dramatic enhancement of calcium DS (DSI=66%) in this condition compared to an identical model trained on 5 velocities (DSI=17%). Thus, evolutionary search is capable of finding the threshold-based solution, but only when the training is performed on a single stimulus velocity (Figure 2-S2). This solution did not generalize to multiple stimuli speeds because, as mentioned above, they lead to different postsynaptic depolarization levels (Figure 2, 2-S1). Instead, the algorithm converged on a set of postsynaptic paraments leading to less nonlinear calcium channel activation over a broader voltage range, ensuring effective DS performance over multiple velocities and heterogenous local potentials (Wu et al., 2023).

2. Functionally distinct responses across different regions of interest (ROIs) were used to classify BC input. ROIs were obtained from multiple scan fields and retinas and combined into a single dataset for functional clustering. However, the consistency of the cluster distribution across these replicates has not been addressed. As BCs can exhibit different functional properties dependant on the state/health of the retina, it is important to know whether certain functional clusters may originate disproportionately from a particular experiment, as it implies that each cluster does not represent a different stable functional/anatomical population.

We acknowledge that the state of the preparation can significantly impact signal dynamics. In response to this important consideration, we have incorporated details about the distribution of functional clusters in various experiments in the revised version of the manuscript (Figure 6-S1, and discussion).

Other comments:3. Interpreting iGluSnFR signals: Since the sensor is expressed uniformly across the SAC dendrite, it is important to clarify why the measured F signals are considered synaptic responses. Could spillover contribute to the generation of slower responses?

We do not believe spillover can explain slower responses because the sluggish clusters often responded significantly (up to 500ms) sooner to moving bars (Figures 6, 6-S3). We acknowledge and discuss this possibility of spillover in the revised discussion.

4. One striking finding is the diversity of BCs RF sizes (Fig. 7C). Some BCs have RF that are far larger than their dendritic fields. It will be useful to discuss the potential mechanisms that may underlie large BC RFs.

We changed the discussion to address this question.

5. SAC DS is independent of dendritic isolation: The authors claim that dendritic isolation does not significantly impact DS. However, while this might be true for a linear motion through the receptive field, dendritic isolation probably matters for more dynamic stimuli. For example, DSGCs can encode rapid changes in objection direction, as DS is computed over fine spatiotemporal scales relying on SACs (Murphy-Baum et al., 2022). This could not occur if SAC dendrites were not well electrically isolated from each other.

We believe that this is an accurate interpretation of our findings. Our research suggests that dendritic isolation is likely not a critical factor in the space-time wiring mechanism. However, as we demonstrate that this particular mechanism cannot fully account for the observed levels of DS in SACs, other mechanisms must be important. As previous studies revealed that dendritic isolation enhances SAC DS (for example, Koren et al., 2017), dendritic independence likely contributes to directional performance within SACs by these additional mechanisms.

6. Figure 4: From what I understand, the BC inputs for the electrotonic connectivity variations evolved much like they were for the original model without axial resistance constraints. This makes sense, since stronger/weaker inputs with different temporal kernels may be appropriate for each condition, hence why the axial resistance wasn't changed post-evolution, which would have likely caused the DS to drop. If that is the case, however, I wonder how the best DS attainable by the final model which is constrained to the radial arrangement of realistic BC inputs (without being able to fit much more optimal sustained-transient BCs to their circumstance) would be impacted. Is dendritic isolation similarly unimportant when the pre-synaptic story isn't ideal?

We have explored this question directly by allowing the evolutionary algorithm to modify the passive and active characteristics of the postsynaptic SAC. Our findings are summarized in Figure 9-S1. We observed a correlation between DSI levels and membrane/axial resistance values in SACs in the evolved models. Better DS was seen with leaky membranes (higher isolation) and lower axial resistance (lower isolation). While it is clear that postsynaptic parameters can influence synaptic integration, they can not fully compensate for inadequate presynaptic dynamics.

7. BC are shown to contribute to DS across velocities (Fig. 9), which contrasts with results from Srivastava et al., (2022) that showed BCs contribute to DS at lower velocities. However, this discrepancy can easily be explained by the choice of moving spots. In this study, the sweeping bars had dynamic width (targeting pixel dwell time of 2s), which means for higher velocities the bar is significantly wider. While in the previous study, the width of the stimulus was kept constant, and thus for higher velocities, the sustained/transient kinetic differences of BCs are less clear (Srivastava et al., 2021). The author's should discuss this explicitly, to avoid discrepancies between these two studies the reader might otherwise perceive.

We value reveiwer’s feedback, and in response, we have included an additional paragraph in the manuscript addressing the distinctions in directional tuning that arise from the space-time model presented in this work, in comparison to earlier studies.

8. Methods: It will be good to discuss how ROIs sizes and positions were selected (pixel correlations?)

We have included a more detailed explanation of the clustering procedure

Lines 614 describe whole-cell patch clamp techniques, which are not used in this study.

We used patch-clamp to record the waveforms shown in Figure 2-S2

9. Figure 6: Diversity of Glut responses to motion in ON and OFF SACs, caption typos?"Left:" without "Right:" to describe the population (I presume) viewed as an imageIf there should still be A,C and B,D to group the ON and OFF halves, maybe it should be mentioned in the caption

Thank you for bringing this to our attention, the legends were fixed.

References:

Kim, J. S., Greene, M. J., Zlateski, A., Lee, K., Richardson, M., Turaga, S. C., Purcaro, M., Balkam, M., Robinson, A., Behabadi, B. F., Campos, M., Denk, W., Seung, H. S., & EyeWirers (2014). Space-time wiring specificity supports direction selectivity in the retina. Nature, 509(7500), 331-336. https://doi.org/10.1038/nature13240

Gaynes, J. A., Budoff, S. A., Grybko, M. J., Hunt, J. B., & Poleg-Polsky, A. (2022). Classical center-surround receptive fields facilitate novel object detection in retinal bipolar cells. Nature communications, 13(1), 5575. https://doi.org/10.1038/s41467-022-32761-8

Murphy-Baum B. and Awatramani GB (2022). Parallel processing in active dendrites during periods of intense spiking activity, Cell Reports, Volume 38, Issue 8,

Srivastava P, de Rosenroll G., MatsumotoA., Michaels T., Turple Z., Jain V, Sethuramanujam S, Murphy-Baum B, Yonehara K., Awatramani, G.B. (2022) Spatiotemporal properties of glutamate input support direction selectivity in the dendrites of retinal starburst amacrine cells eLife 11:e81533

Strauss, S., Korympidou, M. M., Ran, Y., Franke, K., Schubert, T., Baden, T., Berens, P., Euler, T., & Vlasits, A. L. (2022). Center-surround interactions underlie bipolar cell motion sensitivity in the mouse retina. Nature communications, 13(1), 5574. https://doi.org/10.1038/s41467-022-32762-7

Tukker, J. J., Taylor, W. R., & Smith, R. G. (2004). Direction selectivity in a model of the starburst amacrine cell. Visual neuroscience, 21(4), 611-625. https://doi.org/10.1017/S0952523804214109

**Reviewer #2 (Recommendations For The Authors):**
Specific comments1. Line 223. The statement a model trained on only optimal DSI would produce "negligible absolute differences in calcium levels." is unclear. This needs to be better explained.

We have modified and expanded this paragraph to make it more clear

2. Figure 4. The authors use this model to test the hypothesis that space time wiring contribution to SAC process DS requires dendritic isolation. They do this by increasing axial resistance around the soma of their model neuron to isolate each dendrite. They found comparable DS was achieved in both conditions, indicating that the space-time wiring model works in two cases of high and low dendritic isolation. However, to test the claim that "specific details of postsynaptic integration appear to play a lesser role" (line 274) the authors may consider allowing the axial resistance to change as a part of the model rather than testing two extreme states.

Membrane and axial resistances (and active parameters) were allowed to change as part of model evolution in most simulations presented in this manuscript. We have added the information on the final resistance values reached in the evolved models in Figure 9-S1

3. Figure 6: To study glutamatergic input onto SACs, the authors expressed iGLuSnFR in ChAT-Cre mice and grouped similarly responding pixels into ROIs and separated these responses into functional groups based on cluster analysis (Figure 5). The alignment of the responses in Figure 6A was confusing. It appears that average responses for each cluster are aligned based on the peak observed during the stimulus in each direction, but it is unclear how they are aligned relative to each other or what this timing is relative to location of the stimulus (i.e. what is time 0 in 6A?).

The displayed traces represent the average responses to horizontally moving bars (speed = 0.5mm/s), either moving to the left or right. To achieve this alignment, we employed a procedure consistent with our recent publication (Gaynes et al., 2022), which we have now detailed more comprehensively. Here's the step-by-step process we followed:

1. Determination of half-maximum rise times: Initially, we calculated the half-maximum rise times for glutamate signals recorded in response to left and right-moving stimuli.

2. Calculation of mean rise time: We then computed the mean of these rise times, which served as a reference point for alignment.

3. Alignment procedure: To illustrate the alignment process, consider an example. Suppose the 50% rise time for responses to left-moving stimuli occurs at 3 seconds, while responses to right-moving stimuli occur 4 seconds after stimulation onset. This discrepancy suggests that the RF of the cell is shifted to the right from the center of the display (assuming a stimulation speed of 0.5mm/s on the retina, the RF's position would be approximately 250μm from the midline). To align these responses, we shifted both waveforms by 500ms so that their 50% rise times coincided at 3.5 seconds. Importantly, 3.5 seconds would represent the 50% rise time of the ROI if it were precisely centered on the display. This alignment effectively removed any spatial position dependence from the ROIs.

4. Comparative analysis and clustering: With the responses now aligned, we were able to compare their shapes and subsequently cluster the ROIs into distinct functional clusters.For clarity, we opted to highlight the time of response peak for cluster 1. Although this peak closely aligned with the calculated time of stimulus motion over the center of the 'shifted RF' in the adjusted time frame, it provided a more straightforward comparison between response dynamics.

4. The authors need to do a better job explaining how their results differ from Ezra-Tsur et al 2021, which uses the same sort of model to address the same question. The discussion about this study (lines 425-435) are based on how a more constrained version of these models work better but they do not directly address the difference in conclusion with regards to mechanisms that contribute to SAC process direction selectivity.

We have expanded the discussion related to mechanisms that contribute to DS in SACs and discuss the differences between our studies.

Minor point: The authors use the word "probe" to refer to visual stimulus. This is confusing because "probe" is also used to refer to sensors.

In the revised manuscript, we minimized the usage of ‘probe’ to reference visual stimuli

**Reviewer #3 (Recommendations For The Authors):**
Writing and figure presentations are excellent.

Thank you!

References:

Franke, K., Berens, P., Schubert, T., Bethge, M., Euler, T., & Baden, T. (2017). Inhibition decorrelates visual feature representations in the inner retina. Nature, 542(7642), 439-444. https://doi.org/10.1038/nature21394

Gaynes, J. A., Budoff, S. A., Grybko, M. J., Hunt, J. B., & Poleg-Polsky, A. (2022). Classical Center-Surround Receptive Fields Facilitate Novel Object Detection in Retinal Bipolar Cells. Nat Commun, 13(1), 5575. https://doi.org/https://doi.org/10.1038/s41467-022-32761-8

Kim, J. S., Greene, M. J., Zlateski, A., Lee, K., Richardson, M., Turaga, S. C., Purcaro, M., Balkam, M., Robinson, A., Behabadi, B. F., Campos, M., Denk, W., Seung, H. S., & EyeWirers. (2014). Space-time wiring specificity supports direction selectivity in the retina. Nature, 509(7500), 331-336. https://doi.org/10.1038/nature13240

Matsumoto, A., Agbariah, W., Nolte, S. S., Andrawos, R., Levi, H., Sabbah, S., & Yonehara, K. (2021). Direction selectivity in retinal bipolar cell axon terminals. Neuron. https://doi.org/10.1016/j.neuron.2021.07.008

Matsumoto, A., Briggman, K. L., & Yonehara, K. (2019). Spatiotemporally Asymmetric Excitation Supports Mammalian Retinal Motion Sensitivity. Curr Biol. https://doi.org/10.1016/j.cub.2019.08.048

Srivastava, P., de Rosenroll, G., Matsumoto, A., Michaels, T., Turple, Z., Jain, V., Sethuramanujam, S., Murphy-Baum, B. L., Yonehara, K., & Awatramani, G. B. (2022). Spatiotemporal properties of glutamate input support direction selectivity in the dendrites of retinal starburst amacrine cells. Elife, 11. https://doi.org/10.7554/eLife.81533

Strauss, S., Korympidou, M. M., Ran, Y., Franke, K., Schubert, T., Baden, T., Berens, P., Euler, T., & Vlasits, A. L. (2022). Center-surround interactions underlie bipolar cell motion sensing in the mouse retina. Nat Commun, 13(1), 5574. https://doi.org/https://doi.org/10.1038/s41467-022-32762-7

Tukker, J. J., Taylor, W. R., & Smith, R. G. (2004). Direction selectivity in a model of the starburst amacrine cell. Vis Neurosci, 21(4), 611-625. http://www.ncbi.nlm.nih.gov/entrez/query.fcgi?cmd=Retrieve&db=PubMed&dopt=Citation&list_uids=15579224

Wu, J., Kim, Y. J., Dacey, D. M., Troy, J. B., & Smith, R. G. (2023). Two mechanisms for direction selectivity in a model of the primate starburst amacrine cell. Vis Neurosci, 40, E003. https://doi.org/10.1017/S0952523823000019